# IGTO: Individual Global Transform Optimization for Multi-Agent Reinforcement Learning

## Abstract

The rigorous equivalency of individual-global actions is accustomedly assumed for Centralized Training with Decentralized Execution (CTDE) in Multi-Agent Reinforcement Learning (MARL), wherever Individual-Global-Max (IGM) or Individual-Global-Optimal (IGO) it is. To relax the restriction, in this work, we pose an individual-global action-transformed condition, named Individual-Global-Transform-Optimal (IGTO), to permit inconsistent individual-global actions while guaranteeing the equivalency of their policy distributions. Conditioned by IGTO, accordingly, we design a Individual-Global Normalized Transformation (IGNT) rule, which could be seamlessly implanted into many existing CTDE-based algorithms. Theoretically, we prove that individual-global policies can converge to the optimum under this rule. Empirically, we integrate IGNT into Multi-agent Actor-Critic (named IGNT-MAC) as well as various MARL algorithms, then test on StarCraft Multi-Agent Challenge (SMAC) and Multi-Agent Particle Environment (MPE). Extensive experiments demonstrate that our method can achieve remarkable improvement over the existing MARL baselines.

## 1 Introduction

In the past few years, multi-agent reinforcement learning (MARL) have undergone broad advances for solving cooperative or competitive multi-agent continuous control tasks, such as real-time strategy games OpenAI et al. (2019); Samvelyan et al. (2019); Kurach et al. (2020); Ye et al. (2020), traffic control Shamsoshoara et al. (2019); Wu et al. (2020); Zimmer et al. (2021); Sheng et al. (2022) and autonomous driving Zhou et al. (2020) etc. However, widespread adoption of cooperative MARL methods in real-world settings has been hampered by two major difficulties. First, the environment becomes non-stationary from the perspective of each individual agent, which results from each agent of team often makes decision based on its own local observation or communication. Second, cooperative MARL approaches are notoriously expensive in terms of their computation complexity of joint action value function, which caused by exponential growth of global state and action space with the increasing number of agents. Both of these issues severely limit the applicability of cooperative MARL to real-world multi-agent problems.

To address the above challenges, the prevailing fashion employed in MARL is centralized training with decentralized execution (CTDE) Oliehoek & Amato (2016) to ensure stable training while enabling agents execute actions in a decentralized manner. Among these CTDE methods, decomposition-based MARL (Decom-MARL) Sunehag et al. (2018); Rashid et al. (2018); Son et al. (2019); Wang et al. (2020; 2021); Hu et al. (2023) has shown particular promise due to its superior performances Sun et al. (2021), which factorize the joint action value function into individual action value functions for any factorizable MARL problems to satisfy Individual-Global-Max (IGM) Son et al. (2019) condition i.e., the optimal consistency between global and individual policies. Unfortunately, the operator $\arg\max$ of the IGM condition limits Decom-MARL methods only applicable to discrete action space. Therefore, FOP Zhang et al. (2021) introduce a more general constraint called Individual-Global-Optimal (IGO) from the perspective of policy consistency by factorizing joint policy into individual policies. Although the IGO constraint extends the factorizable task to both discrete and continuous action spaces, it still strictly requires the optimal joint actions to be consistent with the optimal individual behaviors, which may leads to unsatisfied performance in some complicated environments. More discussions about MARL methods based on the IGM or IGO condition and related works are deferred to Appendix A.

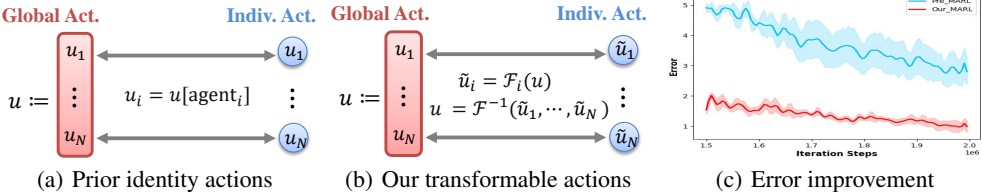

(a) Prior identity actions   (b) Our transformable actions   (c) Error improvement

Figure 1: An illustration of our motivation. (a) Prior methods impose the consistency constraint with identical actions (through direct decomposition or combination) between global and individual learning in CTDE. (b) Our method relaxes this constraint to permit the inconsistency of individual-global actions, through learning some proper transform $\mathcal{F}$. (c) Under the relaxation, the discrepancy between the global Q-value and the combined individual Q-value is reduced sharply for a previous MARL method based on IGM condition, such as QTRAN Son et al. (2019) and the variant implanted with individual-global normalized transformation rule on the 3s_vs_5z scenario of SMAC Samvelyan et al. (2019). More experimental detail and other results are deferred to Section 5 and Appendix G.

In order to relax the restriction, in this paper, we first pose an individual-global action-transformed condition called Individual-Global-Transform-Optimal (IGTO) to permit inconsistent individual-global actions while guaranteeing the equivalency of their policy distributions, as shown in Figure 1(b). In particular, the IGTO condition sequentially performs a series of invertible transformations, which only guarantee a requirement that *the Jacobian determinant of transformation should be equal to 1* i.e., $|\mathbf{G}_i| = 1$. In order to satisfy the requirement, we design an Individual-Global Normalized Transformation (IGNT) rule that map a sample from a simple density i.e., Guassian policy, to a more complex density via the change of variable formula Rudin (1987); Bogachev & Ruas (2007). Furthermore, the IGNT rule can be seamlessly implanted into many existing CTDE-based algorithms. Theoretically, we present and analyze the convergence of individual-global policies under this rule in general cooperative MARL settings. Empirically, we integrate IGNT into Multi-agent Actor-Critic (named IGNT-MAC) as well as various MARL algorithms, then investigate IGNT-MAC in StarCraft Multi-Agent Challenge (SMAC) Samvelyan et al. (2019) and Multi-Agent Particle Environment (MPE) Lowe et al. (2017). In our experiments, we combine IGNT-MAC with four representative CTDE-based MARL methods i.e., VDN Sunehag et al. (2018) for additive value decomposition, QMIX Rashid et al. (2018) for monotonic value decomposition, QTRAN Son et al. (2019) for non-linear value decomposition and FOP Zhang et al. (2021) for the linear value decomposition with entropy regularization. The experimental results demonstrate that the IGNT-MAC induces better performance in terms of convergence speed and stability.

In summary, our contributions are as follows:

- We propose a new condition called **Individual-global-Transform-Optimal (IGTO)** with well theoretical guarantee, which relaxes the constraint of rigorous equivalency of individual-global actions in CTDE-based multi-agent reinforcement learning.

- We design an **Individual-Global Normalized Transformation (IGNT)** rule to satisfy this IGTO condition, which has also theoretical guarantee of optimal individual-global policies.

- We propose an **Individual-Global Normalized Transformation Multi-agent Actor-Critic (IGNT-MAC)** framework integrating IGNT rule for cooperative multi-agent tasks.

- We conduct experiments on SMAC and MPE benchmarks, and the results demonstrate that our IGNT-MAC framework indeed induces better performance and better stability in most tasks, which verifies the benefits of IGNT-MAC in cooperative MARL.

## 2 PRELIMINARIES

### 2.1 PROBLEM FORMULATION

The cooperative MARL problem can be formulated as a decentralized partially observable markov decision process (Dec-POMDP) Oliehoek & Amato (2016), which is defined by a tuple $G = <\mathcal{I}, \mathcal{S}, \mathcal{O}, \mathcal{U}, \mathcal{T}, \mathcal{P}, r, \gamma>$. $\mathcal{I}$ is the team of $N$ agents $\{1, 2, \cdots, N\}$, and $\mathcal{S}$ is a finite set of environmental state. The joint observation at time $t$ is denoted as $o^t = \{o_1^t, \cdots, o_N^t\}$, which consists

of local observation of each agent. At time $t$, each agent takes individual action $u_i^t \in \mathcal{U}_i$ based on its own observation $o_i^t$, and then all individual actions together forming the joint action $u^t \in \mathcal{U}$. Each agent has an action-observation historical trajectory $\tau_i^t = \{o_i^1, u_i^1, \cdots, o_i^{t-1}, u_i^{t-1}, o_i^t\} \in \mathcal{T}^i \equiv (\mathcal{O}_i \times \mathcal{U}_i)^*$, on which it conditions a stochastic policy $\pi_i(u_i|\tau_i) : \mathcal{T} \times \mathcal{U} \to [0, 1]$. The state transition function $\mathcal{P}(s'|s, u) : \mathcal{S} \times \mathcal{U} \times \mathcal{S} \to [0, 1]$ denotes the probability density of the next state $s' \in \mathcal{S}$ given the current state $s \in \mathcal{S}$ and the joint action $u \in \mathcal{U}$. Moreover, the environment generates an immediately global reward $r : \mathcal{S} \times \mathcal{U} \to [r_{min}, r_{max}]$ shared with all agents on each transition, which we will abbreviate as $r \triangleq r(s, u)$ to simplify notation, and $\gamma \in (0, 1]$ is the discount factor that represents the value of time. To address the problem of partial observability that each agent can only obtain its own local observation, the state $s$ is often replaced by the action-observation historical trajectory $\tau$. In addition, we will use $\rho_{\pi_{\text{jt}}}(\tau_t, u_t)$ to denote the state-action marginals of the trajectory distribution induced by joint policy $\pi_{\text{jt}}(u|\tau)$.

## 2.2 MAXIMUM ENTROPY MULTI-AGENT REINFORCEMENT LEARNING

For customary MARL, the goal is to find the optimal joint policy that maximizes the average cumulative reward $\mathbb{E}_{(\tau, u) \sim \rho_{\pi_{\text{jt}}}}[\sum_t \gamma^t r(\tau_t, u_t)]$. To improve learning speed and exploration efficiency Fox et al. (2015); Grau-Moya et al. (2016); Haarnoja et al. (2017); Schulman et al. (2017); Haarnoja et al. (2018), we consider a general maximum entropy MARL(MaxEnt MARL), which augments the reward with an entropy of policy term Ziebart (2010) $\mathcal{H}(\pi(\cdot|\tau)) = -\mathbb{E}[\log\pi(\cdot|\tau)]$. That is, the optimal joint policy aims to maximize its cumulative reward and entropy simultaneously at each visited state, formally,

$$\pi_{\text{jt}}^* = \arg\max_{\pi_{\text{jt}}} \sum_t \mathbb{E}_{(\tau_t, u_t) \sim \pi_{\text{jt}}} [r(\tau_t, u_t) + \alpha\mathcal{H}(\pi_{\text{jt}}(\cdot|\tau_t))], \tag{1}$$

where $\alpha$ is a temperature parameter that determines the relative importance between the entropy term and the global reward. Under the MaxEnt MARL framework, we introduce two value functions: soft joint Q-function $Q_{\text{jt}}^{\text{soft}}(\tau, u)$ and soft joint V-function $V_{\text{jt}}^{\text{soft}}(\tau)$, which are defined similarly for single agent in soft Q-learning Haarnoja et al. (2017). For all agents behaving based on a stochastic joint policy $\rho_{\text{jt}}$, soft joint Q-function denotes the value of joint action-observation historical trajectory-action pair $(\tau_t, u_t)$, which is defined as $Q_{\text{jt}}^{\text{soft}}(\tau_t, u_t) = r(\tau_t, u_t) + \mathbb{E}_{(\tau_{t+1}, \cdots) \sim \rho_{\pi_{jt}}}\left[\sum_{l=1}^{\infty} \gamma^l (r(\tau_{t+l}, u_{t+l}) + \alpha\mathcal{H}(\pi_{\text{jt}}^*(\cdot|\tau_{t+l})))\right]$. Soft joint V-function represents the value of joint action-observation historical trajectory $\tau_t$, which is defined as $V_{\text{jt}}^{\text{soft}}(\tau_t) = \alpha\log \int_{\mathcal{U}} \exp(\frac{1}{\alpha}Q_{\text{jt}}^{\text{soft}}(\tau_t, u'))du'$.

Furthermore, from the perspective of dueling structure $Q = V + A$ proposed by Dueling DQN Wang et al. (2016), we also define a significant quantity, the soft joint A-function $A_{\text{jt}}^{\text{soft}}(\tau, u)$ that denotes the value of joint action $u$ according to a stochastic joint policy $\rho_{\text{jt}}$, which is defined as $A_{\text{jt}}^{\text{soft}}(\tau_t, u_t) = Q_{\text{jt}}^{\text{soft}}(\tau_t, u_t) - V_{\text{jt}}^{\text{soft}}(\tau_t)$. Intuitively, the soft joint A-function denotes the difference between $Q_{\text{jt}}^{\text{soft}}$ and $V_{\text{jt}}^{\text{soft}}$, which implies a relative measure of the importance of joint action $u$.

According to the above definitions, the soft joint Q-function and soft joint V-function satisfy the soft Bellman equation $Q_{\text{jt}}^{\text{soft}}(\tau_t, u_t) = \mathbb{E}_{(\tau_t, u_t) \sim \rho_{\pi_{\text{jt}}}}\left[r(\tau_t, u_t) + \gamma V_{\text{jt}}^{\text{soft}}(\tau_{t+1})\right]$ and $V_{\text{jt}}^{\text{soft}}(\tau_t) = \mathbb{E}_{u_t \sim \pi_{\text{jt}}(\cdot|\tau_t)}\left[Q_{\text{jt}}^{\text{soft}}(\tau_t, u_t) - \alpha\log\pi_{\text{jt}}(\cdot|\tau_t)\right]$. In addition, we can deduce the optimal joint policy $\pi_{\text{jt}}^* = \exp(\frac{1}{\alpha}A_{\text{jt}}^{\text{soft}}(\tau_t, u_t))$. The detailed proof of above consequences are referred to Appendix B.

## 2.3 DECOMPOSITION-BASED MULTI-AGENT REINFORCEMENT LEARNING

In realistic multi-agent environments, each agent only captures local observation because of the limitation of environment, which leads to the decentralized policy may not converge to a global optimal solution. That is, the inconsistency between the optimal joint actions and the optimal individual behaviors often occurs if the agent cannot obtain global information (including other agents' states), which might come from signal interruption, communication limitation or other factors such as in underwater, war battle environments. Therefore, the fashion of centralized training with decentralized execution (CTDE) is most commonly used in cooperative MARL, where each agent learns the

policy by optimizing individual action value function based on the partial observation of each agent and global state during the training phase, and agent makes a decision with local observation at execution phase. In recent years, numerous decomposition-based multi-agent reinforcement learning (Decom-MARL) methods Sunehag et al. (2018); Rashid et al. (2018); Son et al. (2019); Wang et al. (2020; 2021); Shen et al. (2022) have been proposed for the paradigm of CTDE. The key idea of Decom-MARL is to factorize the joint action value function into individual action value function for any factorizable MARL tasks to make sure the consistency between global and individual optimal policy, which can be formulated as follows,

$$Q_{\text{jt}}(\tau, u) = F(Q_1(\tau_1, u_1), Q_2(\tau_2, u_2), \cdots, Q_n(\tau_n, u_n)), \tag{2}$$

where $F$ is a decomposition function, $Q_{\text{jt}}(\tau, u)$ denotes the joint action value of joint action-observation historical trajectory $\tau$ and action $u$, and $Q_i(\tau_i, u_i), i = 1, 2, \cdots, n$ represents the individual action value of individual action-observation historical trajectory $\tau_i$ and action $u_i$.

In particular, the decomposition function of Decom-MARL methods roughly fall into two categories: monotonic linear function Sunehag et al. (2018); Rashid et al. (2018); Wang et al. (2020) and nonlinear function Son et al. (2019); Wang et al. (2021); Shen et al. (2022). As representative works, QMIX Rashid et al. (2018) and QTRAN Sunehag et al. (2018) learn a value decomposition by using a monotonic linear function and a nonlinear function, respectively. Most Decom-MARL methods under the CTDE paradigm are required to satisfy **Individual-Global-Max** (IGM) Son et al. (2019), which is a significant condition to keep the consistency between global and individual policy in the centralized training process, formally,

$$\arg\max_u Q_{\text{jt}}(\tau, u) = \left( \arg\max_{u_1} Q_1(\tau_1, u_1), \cdots, \arg\max_{u_N} Q_N(\tau_N, u_N) \right). \tag{3}$$

As also discussed in Wan et al. (2022), although these approaches guarantee the IGM condition, it is impractical for them to learn complete factorized representations in some complicated MARL tasks because the joint action space grows exponentially with the number of agents. Furthermore, the operator $\arg\max$ of IGM condition makes these methods only applicable to discrete action space. Therefore, FOP Zhang et al. (2021) proposes a method that directly factorize the optimal joint policy instead of the joint value function into individual policies, and give a constraint of optimal policy consistency: **Individual-Global-Optimal** (IGO), i.e.,

$$\pi_{\text{jt}}^*(u|\tau) = \prod_{i=1}^N \pi_i^*(u_i|\tau_i), \tag{4}$$

where $\pi_{\text{jt}}^*(u|\tau)$ denotes the optimal joint policy under the global action-observation historical trajectory $\tau$ and $\pi_i^*(u_i|\tau_i)$ is the optimal individual policy under the local action-observation historical trajectory $\tau_i$. Although the IGO condition offers greater generality compared to the IGM condition for any factorizable cooperative MARL tasks and allows the approach (FOP) applied to assignments involving discrete or continuous actions, it learns an optimal joint policy for entropy-regularized Dec-POMDP rather than the original MDP, i.e., the converged policy may be biased Eysenbach & Levine (2019), which might lead to unsatisfied performance in some complicated environments. For example, some experimental results, such as on the scenario 3s_vs_5z and MMM2 of SMAC (Figure 2), show that FOP under the IGO constraint can not achieve satisfactory performance including convergence speed and stability.

## 3 Individual-Global Transform Optimal Condition

In order to relax the restriction of rigorous equivalency of individual-global actions in CTDE-based MARL, in favor of the consistency of policy distribution, we attempt to introduce the normalized transformation constraint between individual and global actions. In specific, considering a category of sequential decision-making problems that can be factorized during the centralized training stage, we define a new condition (**Individual-Global-Transform-Optimal**, IGTO):

**Definition 1 (IGTO).** *For an optimal joint policy $\pi_{jt}^*(u|\tau) : \tau \times u \to [0, 1]$, where $\tau$ is a joint trajectory and $u$ is a joint action, if there exists a transformed joint action $\widetilde{u} =$*

$[\widetilde{u}_1; \widetilde{u}_2; \cdots; \widetilde{u}_N]$, *which is expressed by an invertible transformation $F$, and individual optimal policies $[\pi_i^*(u_i|\tau_i) : \tau_i \times u_i \to [0,1]]_{i=1}^N$, such that the following holds*

$$\pi_{jt}^*(\widetilde{u}|\tau) = \prod_{i=1}^N \widetilde{\pi}_i^*(\widetilde{u}_i|\tau_i) = \prod_{i=1}^N \pi_i^*(u_i|\tau_i), \tag{5}$$

$$s.t. \quad \widetilde{u} = F(u), u = F^{-1}(\widetilde{u}). \tag{6}$$

*then, we say that $[\pi_i]$ satisfy IGTO for $\pi_{jt}$ under $\tau$. That is, $\pi_{jt}^*(u|\tau)$ is factorized by $[\pi_i^*(u_i|\tau_i)]$.*

The individual-global action-transformed condition is permitting inconsistent individual-global actions while guaranteeing the equivalency of their policy distributions. In particular, the requirement of transformation invertibility in the IGTO condition is that we expect to have a well matching relation between individual actions and global behaviors. Furthermore, under the IGTO condition, we can obtain individual transformed policy improvement by the global joint policy optimization in the centralized training procedure, whose proof is deferred to Appendix C.3.

**Theorem 1.** *(Policy Preservation) If we sequentially perform the transformation $f_i$:*

$$[\widetilde{u}_i; \widehat{u}_{-i}] = f_i(u_i, \widehat{u}_{-i}), \quad \widehat{u}_{-i} = [\widetilde{u}_1, \cdots, \widetilde{u}_{i-1}; u_{i+1}, \cdots, u_N], \tag{7}$$

*, the Jacobian matrix of the transformation exists and the Jacobian determinant satisfies $|\boldsymbol{G}_i| = |\frac{\partial f_i}{\partial [u_i; \widehat{u}_{-i}]}| = 1$, then individual global transform optimal in **Definition 1** is provable.*

*Proof.* The proof is deferred to Appendix C.1. □

**Theorem** 1 shows that if and only if the Jacobian determinant satisfies $|\boldsymbol{G}_i| = 1$ for transformation $f_i$, the optimal individual policy distribution will be consistent with the optimal joint policy distribution. Thus, we will introduce how to design the transformation $f_i$ to guarantee this requirement.

## 4 LEARNING TO INDIVIDUAL GLOBAL TRANSFORMATION

In this section, we firstly design a Individual-Global Normalized Transformation (IGNT) rule to satisfy the IGTO constraint. Then we derive the transform policy iteration and prove individual-global policies converge to optimal. Finally we propose a novel cooperative MARL framework, Individual-Global Normalized Transformation Multi-agent Actor-Critic (IGNT-MAC) integrating IGNT. **Note that all proofs (w.r.t Propositon, Lemma and Theory) are deferred to Appendix C.**

### 4.1 INDIVIDUAL-GLOBAL NORMALIZED TRANSFORMATION

In realistic multi-agent settings, the dimensionality of joint action space grows with the increasing number of agents, which results in performing the calculation of the Jacobian determinant $|\boldsymbol{G}_i|$ is computationally demanding. Therefore, we have to carefully design a tractable and flexible transform function. We take advantage of the straightforward observation that *the determinant of a triangular matrix can be efficiently calculated by taking the product of its diagonal elements.* As a result, we build a tractable and flexible transformation called **normalized transformation** by stacking a sequence of bijections. In every bijection, one part of input vector i.e., $u_i$ is updated using a function that is easy to reverse, but its outcome relies on the remaining part of input vector i.e., $[\widetilde{u}_1, \cdots, \widetilde{u}_{i-1}, u_{i+1}, \cdots, u_N]$ in a sophisticated manner.

**Definition 2** (**Normalized Transformation**). *Given the individual actions $[u_i]_{i=1}^N$, and transformed action variables $[\widetilde{u}_i]_{i=1}^N$, the bijection function $f_i(u_i, \widehat{u}_{-i}) : u = [\widetilde{u}_1, \cdots, \widetilde{u}_{i-1}, u_i, \cdots, u_N] \to \widetilde{u} = [\widetilde{u}_1, \cdots, \widetilde{u}_i, u_{i+1}; \cdots; u_N]$ can be defined as*

$$\begin{cases} \widetilde{u}_i = u_i \odot \frac{1}{F_i}\exp(g_i(\widehat{u}_{-i})) + h_i(\widehat{u}_{-i}) \\ \widetilde{u}_{-i} = \widehat{u}_{-i} \end{cases} \tag{8}$$

*where $\widehat{u}_{-i} = [\widetilde{u}_1, \cdots, \widetilde{u}_{i-1}; u_{i+1}, \cdots, u_N]$, $F_i = \exp(\sum g(\widehat{u}_{-i}))$ is a normalization factor, $g_i$ and $h_i$ stand for scale and translation, and are functions from $\mathbb{R}^{N-1} \mapsto \mathbb{R}$, and $\odot$ is the element-wise product or Hadamard product.*

**Proposition 1.** *The transformation in **Definition** 2 makes policy preservation in **Theorem** 1.*

The **Proposition** 1 indicates that for any factorizable MARL tasks, the optimal individual action distributions and the global behavior distribution are aligned by applying a series of reversible affine transformations to joint action. Moreover, the proof of **Proposition** 1 demonstrates that the scale and translation functions $g, h$ can be arbitrarily intricate, because the computation of the Jacobian determinant for the affine transformation operation does not require calculating the Jacobian of $g, h$. Therefore, we will utilize deep convolutional neural networks for them. Note that the hidden layers of both $g$ and $h$ can contain a higher number of features compared to their input and output layers.

## 4.2 Transform Policy Iteration

According to the above introduction of IGTO condition, we can derive transform policy iteration for completing the update of individual-global transformed policy that alternates between transform policy evaluation and transform policy improvement under the maximum entropy MARL framework. To give conveniently a theoretical analysis, the derivation is based on the discrete action setting.

In the policy evaluation step of transform policy iteration, we need to calculate the soft joint value for the joint policy $\pi_{\text{jt}}$ according to the goal of maximum entropy MARL in Eq. (1). We firstly define the transform Bellman backup operator $\mathcal{T}_{\pi_{\text{jt}}}^{\text{tra}}$ as

$$\mathcal{T}_{\pi_{\text{jt}}}^{\text{tra}} Q_{\text{jt}}^{\text{soft}}(\tau_t, \widetilde{u}_t) \triangleq r(\tau_t, \widetilde{u}_t) + \gamma \mathbb{E}_{\tau_{t+1} \sim \rho} \left[ V_{\text{jt}}^{\text{soft}}(\tau_{t+1}) \right], \tag{9}$$

where

$$V_{\text{jt}}^{\text{soft}}(\tau_t) = \mathbb{E}_{\widetilde{u} \sim \pi_{\text{jt}}} \left[ Q_{\text{jt}}^{\text{soft}}(\tau_t, \widetilde{u}_t) - \log \pi_{\text{jt}}(\widetilde{u}_t | \tau_t) \right]. \tag{10}$$

Then we can update the soft joint Q-function $Q_{\text{jt}}^{\text{soft}}$ for a fixed joint policy by repeatedly applying the transform Bellman backup operator.

**Lemma 1** (**Transform Policy Evaluation**). *Consider the transform Bellman backup operator $\mathcal{T}_{\pi_{jt}}^{tra}$ in Eq. (9) and a mapping $Q_{jt}^0 : \tau \times \widetilde{\mathcal{U}} \to \mathbb{R}$ with $|\mathcal{A}| < \infty$, and define $Q_{jt}^{k+1} = \mathcal{T}_{\pi_{jt}}^{tra} Q_{jt}^k$. Then the sequence $Q_{jt}^k$ will converge to the soft joint Q-value of $\pi_{jt}$ as $k \to \infty$.*

In the policy improvement step of transform policy iteration, we update the joint policy to the exponential of the new value function e.g. $\frac{1}{\alpha} A_{\text{jt}}^{\text{soft}}(\tau, u)$. However, it is intractable to perform this form of joint policy precisely in practice. Therefore, we will restrict the joint policy to some set of tractable policies $\Pi$, for example, a parameterized family of distributions. From the definition of IGTO, the joint policy can be represented by the combination of individual policies. In particular, the individual action $\widetilde{u}_i$ is transformed by an affine transformation from the joint action $u$. Thus, we can project the new individual policies to the desired set of policies. To compute conveniently projection, we choose the K-L divergence, and the update rule of joint policy is given by

$$\pi_{\text{jt}}^{\text{new}} = \arg\min_{\pi \in \Pi} D_{\text{KL}}\left(\pi(\cdot | \tau_t) \,\Big\|\, \frac{\exp(Q^{\pi_{\text{jt}}^{\text{old}}}(\tau_t, \cdot))}{Z^{\pi_{\text{jt}}^{\text{old}}}(\tau_t)}\right), \tag{11}$$

where $Z^{\pi_{\text{jt}}^{\text{old}}}(\tau_t)$ is a normalization term. According to the update of joint policy, we will prove that the Q-value of new joint policy is higher than the old joint policy and formalize the following lemma.

**Lemma 2** (**Transform Policy Improvement**). *Let $\pi_{jt}^{old} \in \Pi$ and let $\pi_{jt}^{new}$ be the optimizer of the minimization problem defined in Eq. (11). Then $Q^{\pi_{jt}^{new}}(\tau_t, \widetilde{u}_t) \geq Q^{\pi_{jt}^{old}}(\tau_t, \widetilde{u}_t)$ for all $(\tau_t, \widetilde{u}_t) \in \tau \times \widetilde{\mathcal{U}}$ with $|\widetilde{\mathcal{U}}| < \infty$.*

In particular, lemma 1 and 2 can be regarded as a consequence of the conclusion drawn from SAC Haarnoja et al. (2018). Furthermore, lemma 2 shows that if we have an existing old policy $\pi_{\text{jt}}^{\text{old}}$ and we can find a new policy $\pi_{\text{jt}}^{\text{new}}$ that satisfies Eq. (11), the new policy $\pi_{\text{jt}}^{\text{new}}$ is superior to the old policy $\pi_{\text{jt}}^{\text{old}}$. By alternating iteratively between the transform policy evaluation and the transform policy improvement, the joint policy will provably converge to the optimal solution, which is formalized as Theorem 2.

**Theorem 2** (**Transform Policy Iteration**). *Repeated application of Transform Policy Evaluation and Transform Policy Improvement, we can obtain a sequence $Q_{soft}^k$ and this sequence will converge towards the optimal soft Q-function $Q_{soft}^*$, while the corresponding sequence of policies will converge towards the optimal policy $\pi_{jt}^*$.*

Theorem 2 indicates that by repeatedly applying transform policy evaluation and transform policy improvement, the joint policy can progressively enhance and ultimately converge to the optimal policy. According to the above discussion, we will extend the transform policy iteration into the continuous setting and propose a practical Individual-Global Normalized Transformation Multi-agent Actor-Critic (IGNT-MAC) framework integrating IGNT rule for cooperative multi-agent tasks.

### 4.3 IGNT MULTI-AGENT ACTOR-CRITIC

We first present the practical update rule of the critic of IGNT-MAC. According to the definition of transform bellman backup operator in Eq. (9), the soft joint Q-value can be expressed as $Q_{jt}^{soft}(\tau_t, \widetilde{u}_t) = r(\tau_t, \widetilde{u}_t) + \gamma \mathbb{E}_{\tau_{t+1} \sim \rho, \widetilde{u} \sim \pi_{jt}} \left[ Q_{jt}^{soft}(\tau_{t+1}, \widetilde{u}_{t+1}) - \log \pi_{jt}(\widetilde{u}_{t+1} | \tau_{t+1}) \right]$. Therefore, the critic is trained by minimize the following loss:

$$\mathcal{L}_{Q_{jt}}(\theta) = \sum_{i=1}^{b} \left[ (y_{jt}^i - Q_{jt}^{soft}(\tau, \widetilde{u}; \theta))^2 \right], \tag{12}$$

where $y_{jt} = r(\tau, \widetilde{u}) + \gamma \left[ Q_{jt}^{soft}(\tau', \widetilde{u}'; \overline{\theta}) - \log \pi_{jt}(\overline{u}' | \tau') \right]$, $\theta$ and $\overline{\theta}$ denote the parameters of current and target networks, and $b$ represent the batch size of transitions sample from the replay buffer.

Then, according to the Lemma 2, we can derive the objective of joint policy,

$$J_{\pi_{jt}}(\phi) = \mathbb{E}_{\tau \sim \rho} \left[ D_{\text{KL}}(\pi_{jt}(\cdot | \tau; \phi) \parallel \frac{\exp(Q_{jt}(\tau, \cdot))}{Z_{jt}(\tau)}) \right]. \tag{13}$$

The optimal joint policy can be expressed as follows,

$$
\begin{aligned}
\pi_{jt} &= \arg \min_{\pi_{jt}} D_{\text{KL}}(\pi_{jt}(\cdot | \tau) \parallel \frac{\exp(Q_{jt}(\tau, \cdot))}{Z_{jt}(\tau)}) \\
&= \arg \max_{\pi_{jt}} \sum_{\widetilde{u}} \pi_{jt}(\widetilde{u} | \tau) \left( Q_{jt}(\tau, \widetilde{u}) + \log \frac{Z_{jt}(\tau)}{\pi_{jt}(\widetilde{u} | \tau)} \right).
\end{aligned}
\tag{14}
$$

From the IGTO condition, we can obtain the joint policy $\pi_{jt}(\widetilde{u} | \tau) = \prod_{i=1}^{N} \pi_i(u_i | \tau_i)$. Thus, the objective of each actor (individual policy) $\pi_i$, parameterized by $\phi_i$, can be formulated as follows,

$$J_{\pi_i}(\phi_i) = \mathbb{E}_{\tau_i \sim \rho_i} \left[ \sum_{\widetilde{u}_i} \pi_i(u_i | \tau_i) \left( Q_{jt}(\tau, \widetilde{u}) + \log \frac{Z_{jt}(\tau)}{\pi_i(u_i | \tau_i)} \right) \right], \tag{15}$$

which can be optimized with stochastic gradients,

$$\hat{\nabla}_{\phi_i} \mathcal{J}_{\pi_i}(\phi_i) = \nabla_{\phi_i} \log \pi_i(u_i | \tau_i) \left[ Q_{jt}(\tau, \widetilde{u}) - \log \pi_i(u_i | \tau_i) \right]. \tag{16}$$

Since the objective of each individual policy in Eq. (15) is independent of the behaviors of the other agents, the gradient exhibits lower variance compared to MARL methods that rely on a centralized critic. According to the above update rules of IGNT-MAC, the training process for the IGNT-MAC framework are presented in Appendix D.

## 5 EXPERIMENTS

The goal of our experiments is to explore and examine how IGNT-MAC enhances the performance of existing MARL algorithms. IGNT-MAC is a flexible framework and can be integrated with numerous existing MARL algorithms. In our experiments, we select four representative methods: VDN Sunehag et al. (2018), QMIX Rashid et al. (2018), QTRAN Son et al. (2019), and

FOP Zhang et al. (2021), which are only adjusted in order to adapt these algorithms to the IGNT-MAC framework. We denote the modified methods as VDN+IGNT, QMIX+IGNT, QTRAN+IGNT and FOP+IGNT. The modifications for these algorithms are limited, aiming to preserve the original architecture and ensure a fair demonstration of the enhancements brought by IGNT. The intricate details of adjustments and hyperparameters are deferred to Appendix F. We use StarCraft Multi-Agent Challenge (SMAC) Samvelyan et al. (2019) and Multi-Agent Particle Environment(MPE) Lowe et al. (2017) to experimentally evaluate the performance of IGNT-MAC in discrete or continuous action space. All experiments are conducted on 2.90GHz Intel Core i7-10700 CPU, 64G RAM and NVIDIA GeForce RTX 3090 GPU. Note that all results are based on four training runs with different random seeds in the experiments.

## 5.1 TEST ON STARCRAFT MULTI-AGENT CHALLENGE

We conduct experiments in the three different level tasks including 1c3s5z(easy), 3s_vs_5z(hard) and MMM2(super hard). In SMAC, agents, which select actions that conditions on local observation in limited field of view by a MARL approach, compete against an integrated game AI, striving to defeat their opponents. For all battle scenarios, the goal is to maximize the win rate and episode reward. The more details of SMAC environment and training are introduced in Appendix E.1.

The learning curves of average success rates in three scenarios are illustrated in Figure 2(four columns for four methods and three rows for three various maps). Additionally, the learning curves in terms of average episode rewards are provided in Appendix G including three additional scenarios such as 8m, 2c_vs_64zg and 25m. We can observed that the two performance metrics (average win rates and episode rewards) are not always positively correlated. This is evident in the case of QMIX and QTRAN in the 3s_vs_5z scenario. The reason for this discrepancy is that the primary objective of the task of StarCraft II is to achieve victory by eliminating all enemies. However, MARL algorithms aim to maximize the average reward, which includes both the episodic reward (i.e., game victory) and the immediate rewards (i.e., inflicting damage to enemies or receiving self-damage penalties).

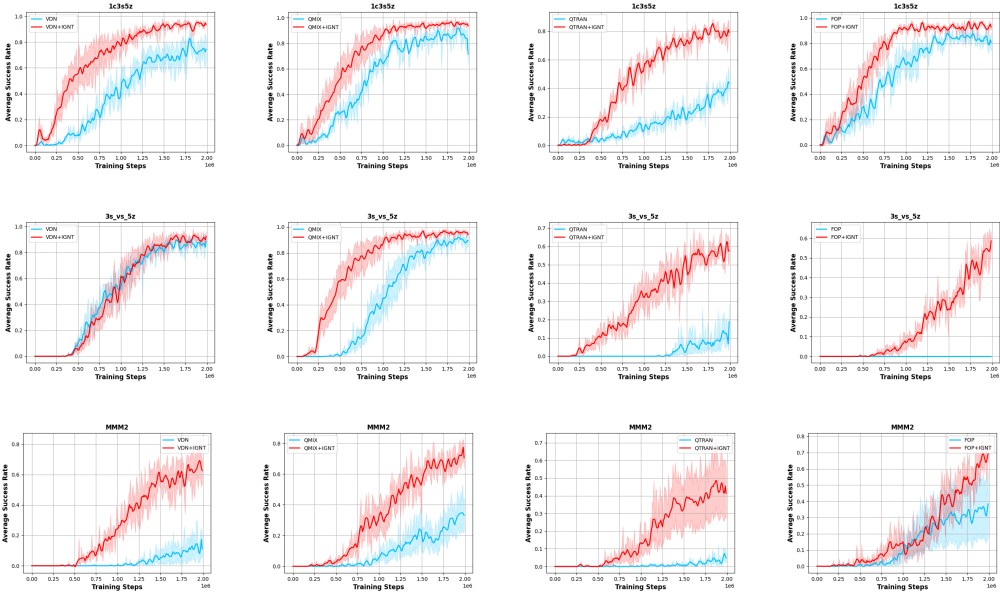

Figure 2: The performance in terms of average success rates for some baselines (VDN, QMIX, QTRAN and FOP) and their variants under the framework of IGNT-MAC on different scenarios of SMAC benchmark including 1c3s5z, 3s_vs_5z and MMM2, which respectively correspond to the three levels of difficulties: easy, hard and super hard. In particular, each row exhibits the results of different groups in the same scenario.

The experimental results demonstrate that, in general, these variants within the IGNT-MAC framework exhibit comparable performance to the baseline methods in the easy scenario, while surpassing them by a significant margin in the more challenging tasks. This superiority is evident in terms of learning speed and final performance. In specific, we can observe that the factorized value-based

methods (VDN, QMIX, QTRAN) and factorized actor-critic approach (FOP) integrated with IGNT rule exhibit fast learning compared to the original baselines in the easy task (1c3s5z), which highlights the efficiency of individual-global normalized transformation. In all scenarios, FOP+IGNT outperforms FOP significantly and exhibits average win rates that are higher than 60% and 30% on the 3s_vs_5z and MMM2. The above experimental results can be an evidence for the advantages of IGNT rule which could ensure the policy monotonic improvement in the Dec-POMDP.

## 5.2 TEST ON MULTI-AGENT PARTICLE ENVIRONMENT

The Multi-agent particle environment(MPE) contains cooperative or competitive games, which consists of $N$ agents, landmarks and obstacles inhabiting a two dimensional world with continuous space and discrete time. Agents can perform physical actions within the environment and communication actions that get broadcasted to other agents. In our experiments, we exclusively consider cooperative tasks to assess the effectiveness of the IGNT-MAC framework. The more details regarding these cooperative tasks are deferred to Appendix E.2.

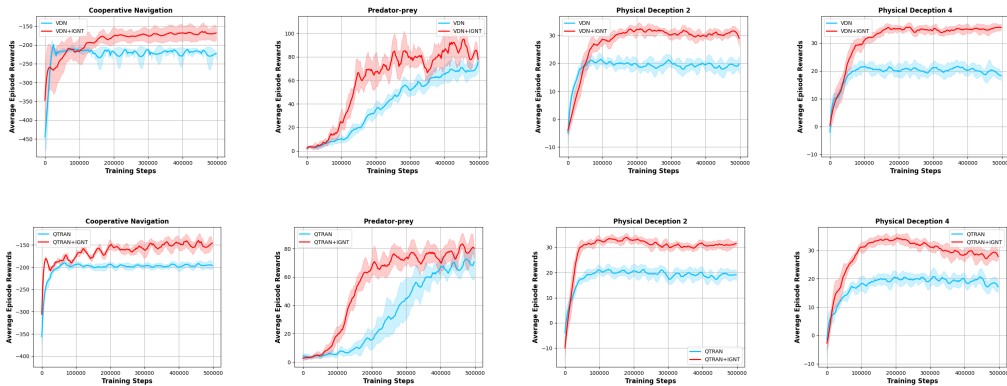

Figure 3: The performance in terms of average episode rewards for some baselines (VDN and QTRAN) and their variants under the framework of IGNT-MAC on different tasks of MPE including Cooperative Navigation, Modified Predator-prey, Physical Deception 2 and Physical Deception 4. In particular, each columns exhibits the results of different groups in the same scenario.

We evaluate the two groups of methods using the MPE benchmark, and their performance is shown in Figure 3. It can be observed that IGNT-MAC outperforms the comparison baselines in all four scenarios at the end of training. Furthermore, VDN+IGNT and QTRAN+IGNT exhibit superior learning rates compared to the original methods. Although some baselines learns faster than the variants under the IGNT-MAC framework in some scenarios such as Physical Deception 2, they tend to converge to suboptimal policies, whereas IGNT-MAC is able to discover better policies to successfully complete the cooperative tasks.

## 6 CONCLUSION

In this paper, we proposed a novel cooperative MARL framework called Individual-Global Normalized Transformation Multi-agent Actor-Critic (IGNT-MAC). In particular, to relax the restriction, i.e., IGM or IGO, for rigorous equivalency of individual-global actions, we posed an individual-global action-transformed condition named individual-global-Transform-Optimal (IGTO) to permit inconsistent individual-global actions while guaranteeing the equivalency of their policy distributions. In order to satisfy the IGTO condition, we designed an reversible Individual-Global Normalized Transformation (IGNT) for global joint actions to make policy preservation, which could be seamlessly implanted into many existing CTDE-based algorithms. Furthermore, we proved theoretically the individual-global transform policies converge to optimum under the IGNT rule. The experimental results in StarCraft Multi-Agent Challenge (SMAC) and Multi-Agent Particle Environment (MPE) demonstrate that the IGNT rule implanted into existing MARL methods outperform the original approaches in terms of convergence speed and stability.

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
