# A   RELATED WORK

Several value-based reinforcement learning methods Sunehag et al. (2018); Rashid et al. (2018); Son et al. (2019); Yang et al. (2020); Rashid et al. (2020); Wang et al. (2021); Sun et al. (2021); Zohar et al. (2022); Shen et al. (2022) are proposed to represent the global action value function with individual counterpart whose complexity grows exponentially with the number of agents, due to partial observability and large joint action. Specifically, VDN Sunehag et al. (2018) and QMIX Rashid et al. (2018) learn a linear value decomposition by additivity and monotonicity. Qatten Yang et al. (2020) is an extensive work of VDN, which uses the multi-head attention mechanism to approximate joint action value. Besides, Weight QMIX Rashid et al. (2020) introduces a weighted projection to rectify the challenge, suboptimal policies, caused by QMIX. QTRAN Son et al. (2019), DFAC Sun et al. (2021) and QPLEX Wang et al. (2021) extend further the action-value function conditions on Individual Global Max(IGM) to keep consistency of global and individual optimal action. However, almost approaches (except QPLEX) are still suffer from the simplicity of decomposition and relaxation of these constraints, which may results in poor performance in some complex tasks. Last but not least, the value-based Decomposition methods are limited to handle discrete action space.

In order to deal with discrete or continuous action space, some policy-based methods Lowe et al. (2017); Foerster et al. (2018); de Witt et al. (2020); Wang et al. (2020); Kim et al. (2021); Zhang et al. (2021) have been developed in recent years. MADDPG Lowe et al. (2017) and COMA Foerster et al. (2018) are the variant of actor-critic method, which learn a centralized critic instead of individual critic based on the observation and action of agent. In particular, Compared with MADDPG, COMA only learn an actor network by sharing parameters to speed learning. de Witt et al. (2020) proposed a novel method called FacMADDPG, which facilitates the critic in decentralized POMDPs based on MADDPG and QMIX. DOP Wang et al. (2020) introduces firstly the value decomposition similar to Qatten Yang et al. (2020) into multi-agent actor-critic framework with on-policy TD($\lambda$) and tree backup technique. Kim et al. (2021) poses a meta-learning multi-agent policy gradient theorem to adapt quickly the non-stationarity of environment, and gives the theoretical analysis in detail. However, due to centralized-decentralized mismatch (CDM) problems, the above methods perform unsatisfactorily compared with value based methods. Furthermore, FOP Zhang et al. (2021) is proposed to solve the above dilemma by transforming IGM into Individual-Global-Optimal (IGO) conditions. However, as pointed out by Eysenbach & Levine (2019), the converged policy of FOP may be biased. Although, FOP proves that factorized individual policies can converge to the global optimum, there is an relatively strict constraint that the optimal joint behavior is required to be consistent with the combination of optimal individual behaviors, which may failing to achieve high performance in some complicated MARL tasks.

# B   MAXIMUM ENTROPY MULTI-AGENT REINFORCEMENT LEARNING PROOF

**Theorem 3** (**Soft Bellman Equation**). *According to the definition of soft joint Q-function $Q_{jt}^{soft}$ and soft joint V-function $V_{jt}^{soft}$, $Q_{jt}^{soft}(\tau, u)$ and $V_{jt}^{soft}(\tau)$ satisfy the soft Bellman equation*

$$Q_{jt}^{soft}(\tau_t, u_t) = \mathbb{E}_{(\tau_t, u_t) \sim \rho_{\pi_{jt}}} \left[ r(\tau_t, u_t) + \gamma V_{jt}^{soft}(\tau_{t+1}) \right], \tag{17}$$

$$V_{jt}^{soft}(\tau_t) = \mathbb{E}_{u_t \sim \pi_{jt}(\cdot|\tau_t)} \left[ Q_{jt}^{soft}(\tau_t, u_t) - \alpha \log \pi_{jt}(\cdot|\tau_t) \right]. \tag{18}$$

*Proof.* Recall the definition of the soft joint V-function:

$$V_{jt}^{\text{soft}}(\tau_t) = \mathbb{E}_{(\tau_{t+l}, \cdots) \sim \rho_{\pi_{jt}}} \left[ \sum_{l=0}^{\infty} \gamma^l (r(\tau_{t+l}, u_{t+l})) + \alpha \mathcal{H}(\pi_{jt}^*(\cdot|\tau_{t+l}))) \right]. \tag{19}$$

Then the soft joint Q-function can be shown that

$$\begin{aligned} Q_{jt}^{\text{soft}}(\tau_t, u_t) &= \mathbb{E}_{(\tau_t, u_t) \sim \rho_{\pi_{jt}}} \left[ r(\tau_t, u_t) + \gamma (Q_{jt}^{\text{soft}}(\tau_{t+1}, u_{t+1}) + \alpha \mathcal{H}(\pi_{jt}(\cdot|\tau_t))) \right] \\ &= \mathbb{E}_{(\tau_t, u_t) \sim \rho_{\pi_{jt}}} \left[ r(\tau_t, u_t) + \gamma V_{jt}^{\text{soft}}(\tau_{t+1}) \right]. \end{aligned} \tag{20}$$

Similarly, we can show that

$$
\begin{aligned}
V_{\mathrm{jt}}^{\mathrm{soft}}(\tau_t) &= \mathbb{E}_{u_t \sim \pi_{\mathrm{jt}}(\cdot|\tau_t)} \left[ Q_{\mathrm{jt}}^{\mathrm{soft}}(\tau_t, u_t) \right] + \alpha \mathcal{H}(\pi_{\mathrm{jt}}(\cdot|\tau_t)) \\
&= \mathbb{E}_{u_t \sim \pi_{\mathrm{jt}}(\cdot|\tau_t)} \left[ Q_{\mathrm{jt}}^{\mathrm{soft}}(\tau_t, u_t) \right] - \alpha \mathbb{E}_{u_t \sim \pi_{\mathrm{jt}}(\cdot|\tau_t)} \left[ \log \pi_{\mathrm{jt}}(\cdot|\tau_t) \right] \\
&= \mathbb{E}_{u_t \sim \pi_{\mathrm{jt}}(\cdot|\tau_t)} \left[ Q_{\mathrm{jt}}^{\mathrm{soft}}(\tau_t, u_t) - \alpha \log \pi_{\mathrm{jt}}(\cdot|\tau_t) \right].
\end{aligned}
\tag{21}
$$

This completes the proof of Theorem 3. □

**Theorem 4** (**Optimal Joint Policy**). *According to the definition of soft joint Q-function $Q_{jt}^{soft}$, soft joint V-function $V_{jt}^{soft}$ and soft joint A-function $A_{jt}^{soft}$, the optimal joint policy is given by*

$$
\pi_{jt}^* = \exp(\frac{1}{\alpha} A_{jt}^{soft}(\tau_t, u_t)).
\tag{22}
$$

*Proof.* From the goal of MaxEnt MARL in Eq. (1), we can derive the objective of discounted maximum entropy multi-agent reinforcement learning:

$$
\mathcal{J}(\pi_{\mathrm{jt}}) = \sum_{t=1}^{T} \gamma^{t-1} \mathbb{E}_{(\tau_t, u_t) \sim \rho_{\pi_{\mathrm{jt}}}} \left[ r(\tau_t, u_t) + \alpha \mathcal{H}(\pi_{\mathrm{jt}}(\cdot|\tau_t)) \right].
\tag{23}
$$

This objective corresponds to maximizing the discounted expected reward and entropy for future action-observation historical trajectory originating from every tuple $(\tau_t, u_t)$ weighted by its probability $\rho_{\pi_{\mathrm{jt}}}$ under the current joint policy. Therefore, the objective for $t = T$ is given by

$$
\begin{aligned}
\mathcal{J}(\pi_{\mathrm{jt}}) &= \mathbb{E}_{u_T \sim \pi_{jt}(\cdot|\tau_T)} \left[ r(\tau_T, u_T) + \alpha \mathcal{H}(\pi_{\mathrm{jt}}(\cdot|\tau_T)) \right] \\
&= \int_{\mathcal{U}} \left[ r(\tau_T, u_T) + \alpha \mathcal{H}(\pi_{\mathrm{jt}}(\cdot|\tau_T)) \right] \pi_{\mathrm{jt}}(u_T|\tau_T) du_T \\
&= \int_{\mathcal{U}} r(\tau_T, u_T) \pi_{jt}(u_T|\tau_T) du_T + \alpha \int_{\mathcal{U}} \pi_{jt}(u_T|\tau_T) \mathbb{E}_{u_T \sim \pi_{jt}(u_T|\tau_T)} \left[ -\log \pi_{jt}(u_T|\tau_T) \right] du_T \\
&= \int_{\mathcal{U}} r(\tau_T, u_T) \pi_{jt}(u_T|\tau_T) du_T + \alpha \mathbb{E}_{u_T \sim \pi_{\mathrm{jt}}(u_T|\tau_T)} \left[ -\log \pi_{\mathrm{jt}}(u_T|\tau_T) \right] \\
&= \int_{\mathcal{U}} r(\tau_T, u_T) \pi_{\mathrm{jt}}(u_T|\tau_T) du_T - \alpha \int_{\mathcal{U}} \log \pi_{\mathrm{jt}}(u_T|\tau_T) \pi_{\mathrm{jt}}(u_T|\tau_T) du_T \\
&= \int_{\mathcal{U}} \left[ r(\tau_T, u_T) - \alpha \log \pi_{\mathrm{jt}}(u_T|\tau_T) \right] \pi_{\mathrm{jt}}(u_T|\tau_T) d\mu_T.
\end{aligned}
\tag{24}
$$

The optimal joint policy at last timestep can be expressed as follows:

$$
\begin{aligned}
\pi_{\mathrm{jt}}^*(\cdot|\tau_T) &= \arg\max_{\pi_{\mathrm{jt}}(\cdot|\tau_T)} \int_{\mathcal{U}} \left[ r(\tau_T, u_T) - \alpha \log \pi_{\mathrm{jt}}(u_T|\tau_T) \right] \pi_{\mathrm{jt}}(u_T|\tau_T) du_T \\
&= \frac{\exp(\frac{1}{\alpha} r(\tau_T, u_T))}{\int_{\mathcal{U}} \exp(\frac{1}{\alpha} r(\tau_T, u)) du} = \exp(\frac{1}{\alpha} A_{\mathrm{jt}}^{\mathrm{soft}}(\tau_T, u_T)).
\end{aligned}
\tag{25}
$$

Similarly, we can show that

$$
\begin{aligned}
\pi_{\mathrm{jt}}^*(\cdot|\tau_t) &= \arg\max_{\pi_{jt}(\cdot|\tau_t)} \mathbb{E}_{u_t \sim \pi_{\mathrm{jt}}(\cdot|\tau_t)} \left[ r(\tau_t, u_t) + \alpha \mathcal{H}(\pi_{\mathrm{jt}}(\cdot|\tau_t)) + \gamma \mathbb{E}_{p(\tau_{t+1}|\tau_t, u_t)} \left[ V_{\mathrm{jt}}^{\mathrm{soft}}(\tau_{t+1}) \right] \right] \\
&= \arg\max_{\pi_{\mathrm{jt}}(\cdot|\tau_t)} \int_{\mathcal{U}} \left[ r(\tau_t, u_t) - \alpha \log(\pi_{\mathrm{jt}}(u_t|\tau_t)) + \gamma \mathbb{E}_{p(\tau_{t+1}|\tau_t, u_t)} \left[ V_{\mathrm{jt}}^{\mathrm{soft}}(\tau_{t+1}) \right] \right] \pi_{\mathrm{jt}}(u_t|\tau_t) du_t \\
&= \frac{\exp\{\frac{1}{\alpha} \left[ r(\tau_t, u_t) + \gamma \mathbb{E}_{p(\tau_{t+1}|\tau_t, u_t)} \left[ V_{\mathrm{jt}}^{\mathrm{soft}}(\tau_{t+1}) \right] \right]\}}{\int_{\mathcal{U}} \exp\{\frac{1}{\alpha} \left[ r(\tau_t, u) + \gamma \mathbb{E}_{p(\tau_{t+1}|\tau_t, u_t)} \left[ V_{\mathrm{jt}}^{\mathrm{soft}}(\tau_{t+1}) \right] \right]\} du} \\
&= \frac{\exp(\frac{1}{\alpha} Q_{\mathrm{jt}}^{\mathrm{soft}}(\tau_t, u_t))}{\int_{\mathcal{U}} \exp(\frac{1}{\alpha} Q_{\mathrm{jt}}^{\mathrm{soft}}(\tau_t, u)) du} = \exp(\frac{1}{\alpha} A_{\mathrm{jt}}^{\mathrm{soft}}(\tau_t, u_t)).
\end{aligned}
\tag{26}
$$

This completes the proof of Theorem 4. □

## C  Individual Global Transform Optimal Proof

**Definition 3** (IGTO). *For an optimal joint policy $\pi_{jt}^*(u|\tau) : \tau \times u \rightarrow [0,1]$, where $\tau$ is a joint trajectory and $u$ is a joint action, if there exists a transformed joint action $\widetilde{u} = [\widetilde{u}_1; \widetilde{u}_2; \cdots; \widetilde{u}_N]$, which is expressed by an invertible transformation $F$, and individual optimal policies $[\pi_i^*(u_i|\tau_i) : \tau_i \times u_i \rightarrow [0,1]]_{i=1}^N$, such that the following holds*

$$\pi_{jt}^*(\widetilde{u}|\tau) = \prod_{i=1}^N \widetilde{\pi}_i^*(\widetilde{u}_i|\tau_i) = \prod_{i=1}^N \pi_i^*(u_i|\tau_i), \tag{27}$$

$$s.t. \quad \widetilde{u} = F(u), u = F^{-1}(\widetilde{u}). \tag{28}$$

*then, we say that $[\pi_i]$ satisfy IGTO for $\pi_{jt}$ under $\tau$. That is, $\pi_{jt}^*(u|\tau)$ is factorized by $[\pi_i^*(u_i|\tau_i)]$.*

### C.1  Policy Preservation for IGTO

**Theorem 5** (**Policy Preservation**). *If we sequentially perform the transformation $f_i$:*

$$[\widetilde{u}_i; \widehat{u}_{-i}] = f_i(u_i, \widehat{u}_{-i}), \quad \widehat{u}_{-i} = [\widetilde{u}_1, \cdots, \widetilde{u}_{i-1}; u_{i+1}, \cdots, u_N], \tag{29}$$

*, the Jacobian matrix of the transformation exists and the Jacobian determinant satisfies $|\mathbf{G}_i| = |\frac{\partial f_i}{\partial [u_i; \widehat{u}_{-i}]}| = 1$, then individual global transform optimal in **Definition 3** is provable.*

*Proof.* For simplification, we denote $p_i = \prod_{k=1}^i \widetilde{\pi}_k(\widetilde{u}_k|\tau_k) \times \prod_{k=i+1}^N \pi_k(u_k|\tau_k)$, where $p_0 = \prod_{k=1}^N \pi_k(u_k|\tau_k)$ and $p_N = \prod_{k=1}^N \widetilde{\pi}_k(\widetilde{u}_k|\tau_k)$. For eq. (27), we only need to prove $p_0 = p_N$. To this end, we take a progressive strategy to derive $p_0 = p_1 = p_2 = \cdots = p_N$. The crucial requirement is $p_{i-1} = p_i$, which is derived as follows.

$$\begin{aligned}
p_{i-1} &= \prod_{k=1}^{i-1} \widetilde{\pi}_k(\widetilde{u}_k|\tau_k) \times \prod_{k=i}^N \pi_k(u_k|\tau_k) \\
&= \prod_{k=1}^{i-1} \widetilde{\pi}_k(\widetilde{u}_k|\tau_k) \times \pi_i(u_i|\tau_i) \times \prod_{k=i+1}^N \pi_k(u_k|\tau_k) \\
&= \pi_i(u_i|\tau_i) \times \pi_{-i}(\widehat{u}_{-i}|\tau_{-i}),
\end{aligned} \tag{30}$$

where $\pi_{-i}(\widehat{u}_{-i}|\tau_{-i}) = \prod_{k=1}^{i-1} \widetilde{\pi}_k(\widetilde{u}_k|\tau_k) \times \prod_{k=i+1}^N \pi_k(u_k|\tau_k)$.

For the individual action $u_i \in \mathcal{U}_i$ and action-observation historical trajectory $\tau_i$, a simple prior probability distribution $\pi_{\widetilde{u}_i}$ on a latent action variable $\widetilde{u}_i \in \widetilde{\mathcal{U}}_i$ under the joint action-observation historical trajectory $\tau$, and a bijection function $f_i : u_i \rightarrow \widetilde{u}_i$ (with $g_i = f_i^{-1}$), **the change of variable formula** defines a policy distribution on $\mathcal{U}_i$ by

$$\pi_i(u_i|\tau_i) = \widetilde{\pi}_i(\widetilde{u}_i|\tau) \times |\mathbf{G}_i|, \tag{31}$$

where $\mathbf{G}_i = \frac{\partial f_i}{\partial [u_i; \widehat{u}_{-i}]}$ is the Jacobian Matrix of $f_i$ at $u_i$.

$$\begin{aligned}
p_{i-1} &= \widetilde{\pi}_i(\widetilde{u}_i|\tau) \times |\mathbf{G}_i| \times \pi_{-i}(\widehat{u}_{-i}|\tau) \\
&= \prod_{k=1}^{i-1} \widetilde{\pi}_k(\widetilde{u}_k|\tau_k) \times \widetilde{\pi}_i(\widetilde{u}_i|\tau) \times |\mathbf{G}_i| \times \prod_{k=i+1}^N \pi_k(u_k|\tau_k) \\
&= \prod_{k=1}^i \widetilde{\pi}_k(\widetilde{u}_k|\tau_k) \times \prod_{k=i+1}^N \pi_k(u_k|\tau_k) \times |\mathbf{G}_i| \\
&= p_i \times |\mathbf{G}_i|.
\end{aligned} \tag{32}$$

As the Jacobian determinant satisfies $|\mathbf{G}_i| = |\frac{\partial f_i}{\partial [u_i; \widehat{u}_{-i}]}| = 1$, the policy transformation has $p_{i-1} = p_i$. Therefore we have proven Theorem 5. $\square$

**Theorem 1** shows that, if the transformation is performed sequentially and the Jacobian determinant of transformation satisfies $|\mathbf{G}_i| = 1$, the IGTO condition will be guaranteed. In fact, the order of performing transformation may be changed. For convenience, we take three agents as example, and change the sequence from the original setting $(f_1, f_2, f_3)$ to a new setting $(f_2, f_3, f_1)$. We summary the conclusion and proof as follows:

**Remark** *Considering a factorizable MARL task with three agents, i.e., $N = 3$. We first perform transformation $f_2$, then transformation $f_3$, finally transformation $f_1$. If the Jacobian determinant of transformation satisfies $|\mathbf{G}_i| = 1, i = 1, 2, 3$, then the IGTO condition is provable.*

**Proof** Let $p_i = \prod_{k=1}^{i} \widetilde{\pi}_k(\widetilde{u}_k|\tau_k) \times \prod_{k=i+1}^{3} \pi_k(u_k|\tau_k)$, where $p_0 = \prod_{k=1}^{3} \pi_k(u_k|\tau_k)$ and $p_3 = \prod_{k=1}^{3} \widetilde{\pi}_k(\widetilde{u}_k|\tau_k)$. For the IGTO condition, we only need to prove $p_0 = p_3$. To this end, we take a progressive strategy to derive $p_0 = p_2 = p_3 = p_1$. From the proof of Theorem 1(Please see **Appendix C.1**), we can obtain the deduction $(\pi_i(u_i|\tau_i) = \widetilde{\pi}_i(\widetilde{u}_i|\tau) \times |\mathbf{G}_i|)$ according to the change of variable formula.

$$
\begin{aligned}
p_0 &= \pi_2(u_2|\tau_2) \times \pi_1(u_1|\tau_1) \times \pi_3(u_3|\tau_3) \\
&= \widetilde{\pi}_2(\widetilde{u}_2|\tau) \times |\mathbf{G}_2| \times \pi_1(u_1|\tau_1) \times \pi_3(u_3|\tau_3) = p_2 \times |\mathbf{G}_2| \\
&= \widetilde{\pi}_3(\widetilde{u}_3|\tau) \times |\mathbf{G}_3| \times \pi_1(u_1|\tau_1) \times \widetilde{\pi}_2(\widetilde{u}_2|\tau) \times |\mathbf{G}_2| = p_3 \times |\mathbf{G}_3| \times |\mathbf{G}_2| \\
&= \widetilde{\pi}_1(\widetilde{u}_1|\tau) \times |\mathbf{G}_1| \times \widetilde{\pi}_2(\widetilde{u}_2|\tau) \times \widetilde{\pi}_3(\widetilde{u}_3|\tau) \times |\mathbf{G}_3| \times |\mathbf{G}_2| \\
&= p_1 \times |\mathbf{G}_1| \times |\mathbf{G}_3| \times |\mathbf{G}_2|
\end{aligned}
\tag{33}
$$

As the Jacobian determinant satisfies $|\mathbf{G}_i| = |\frac{\partial f_i}{\partial [u_i; \widehat{u}_{-i}]}| = 1$, the policy transformation has $p_0 = p_1$. Therefore, we have proven the above theorem.

## C.2 NORMALIZED TRANSFORMATION FOR POLICY PRESERVATION

**Definition 4 (Normalized Transformation).** *Given the individual actions $[u_i]_{i=1}^{N}$, and transformed action variables $[\widetilde{u}_i]_{i=1}^{N}$, the bijection function $f_i(u_i, \widehat{u}_{-i}) : u = [\widetilde{u}_1, \cdots, \widetilde{u}_{i-1}, u_i, \cdots, u_N] \to \widetilde{u} = [\widetilde{u}_1, \cdots, \widetilde{u}_i, u_{i+1}; \cdots ; u_N]$ can be defined as*

$$
\begin{cases}
\widetilde{u}_i = u_i \odot \frac{1}{F_i}\exp(g_i(\widehat{u}_{-i})) + h_i(\widehat{u}_{-i}) \\
\widetilde{u}_{-i} = \widehat{u}_{-i}
\end{cases}
\tag{34}
$$

*where $\widehat{u}_{-i} = [\widetilde{u}_1, \cdots, \widetilde{u}_{i-1}; u_{i+1}, \cdots, u_N]$, $F_i = \exp(\sum g(\widehat{u}_{-i}))$ is a normalization factor, $g_i$ and $h_i$ stand for scale and translation, and are functions from $\mathbb{R}^{N-1} \mapsto \mathbb{R}$, and $\odot$ is the element-wise product or Hadamard product.*

**Proposition 2.** *The transformation in **Definition 4** makes policy preservation in **Theorem** 5.*

*Proof.* According to the **Definition** 4, the Jacobian Matrix of the affine transformation $f_i$ is given by

$$
\mathbf{G}_i = \frac{\partial f_i}{\partial [u_i; \widehat{u}_{-i}]^T} = \begin{bmatrix} \frac{\partial \widetilde{u}_i}{\partial [u_i]^T} & \frac{\partial \widetilde{u}_i}{\partial [\widehat{u}_{-i}]^T} \\ \frac{\partial \widetilde{u}_{-i}}{\partial [u_i]^T} & \frac{\partial \widetilde{u}_{-i}}{\partial [\widehat{u}_{-i}]^T} \end{bmatrix} = \begin{bmatrix} \mathrm{diag}(\frac{1}{F_i}\exp(g_i(\widehat{u}_{-i}))) & \frac{\partial \widetilde{u}_i}{\partial [\widehat{u}_{-i}]^T} \\ 0 & \mathbb{I} \end{bmatrix},
\tag{35}
$$

where $\mathrm{diag}(\frac{1}{F_i}\exp(g_i(\widehat{u}_{-i})))$ is the diagonal matrix whose diagonal elements correspond to the vector $\frac{1}{F_i}\exp(g_i(\widehat{u}_{-i}))$ and $\mathbb{I}$ is an identity matrix. Given the observation that this Jacobian matrix is triangular, we can efficiently calculate its determinant as $|\mathrm{diag}(\frac{1}{F_i}\exp(g_i(\widehat{u}_{-i})))|$. Since $F_i = \exp(\sum g(\widehat{u}_{-i}))$, we can show that

$$
|\mathbf{G}_i| = |\mathrm{diag}(\frac{1}{F_i}\exp(g_i(\widehat{u}_{-i})))| = \frac{1}{F_i} \cdot \exp(\sum g(\widehat{u}_{-i})) = 1.
\tag{36}
$$

As a consequence, the conclusion is reached. $\square$

## C.3 POLICY IMPROVEMENT UNDER IGTO

Under the condition of **IGTO**, the joint policy optimization would improve individual policies. Formally, it can be proved by using the distance measure of KL-divergence, as follows. Let $p_i =$

$\prod_{k=1}^{i} \widetilde{\pi}_k(\widetilde{u}_k|\tau_k) \times \prod_{k=i+1}^{N} \pi_k(u_k|\tau_k)$, where $p_0 = \prod_{k=1}^{N} \pi_k(u_k|\tau_k)$ and $p_N = \prod_{k=1}^{N} \widetilde{\pi}_k(\widetilde{u}_k|\tau_k)$.
According to Eq.( 27), we can show that

$$
\begin{aligned}
D_{KL}(\pi_{jt}(\widetilde{u}|\tau) \parallel \pi_{jt}^*(\widetilde{u}|\tau)) &= D_{KL}(\prod_{i=1}^{N} \widetilde{\pi}_i(\widetilde{u}_i|\tau) \parallel \prod_{i=1}^{N} \pi_i^*(u_i|\tau_i)) \\
&= D_{KL}(p_N \parallel \prod_{i=1}^{N} \pi_i^*(u_i|\tau_i))) \\
&= \int_{\mathcal{U}} p_N \log \frac{p_N}{\prod_{i=1}^{N} \pi_i^*(u_i|\tau_i)} du \\
&= \int_{\mathcal{U}} p_{N-1}|\mathbf{G}_N|^{-1} \log \frac{p_{N-1}|\mathbf{G}_N|^{-1}}{\prod_{i=1}^{N} \pi_i^*(u_i|\tau_i)} du \\
&= \int_{\mathcal{U}} p_{N-2}|\mathbf{G}_{N-1}|^{-1}|\mathbf{G}_N|^{-1} \log \frac{p_{N-2}|\mathbf{G}_{N-1}|^{-1}|\mathbf{G}_N|^{-1}}{\prod_{i=1}^{N} \pi_i^*(u_i|\tau_i)} du \\
&= \cdots = \int_{U} p_0 \prod_{j=1}^{N} |\mathbf{G}_j|^{-1} \log \frac{p_0 \prod_{j=1}^{N} |\mathbf{G}_j|^{-1}}{\prod_{i=1}^{N} \pi_i^*(u_i|\tau_i)} du.
\end{aligned}
\tag{37}
$$

Since each Jacobian determinant $|\mathbf{G}_j| = 1$, then we can show that

$$
\begin{aligned}
D_{KL}(\pi_{jt}(\widetilde{u}|\tau) \parallel \pi_{jt}^*(\widetilde{u}|\tau)) &= \int_{U} p_0 \log \frac{p_0}{\prod_{i=1}^{N} \pi_i^*(u_i|\tau_i)} du \\
&= \int_{U} \prod_{i=1}^{N} \pi_i(u_i|\tau_i) \log \frac{\prod_{i=1}^{N} \pi_i(u_i|\tau_i)}{\prod_{i=1}^{N} \pi_i^*(u_i|\tau_i)} du \\
&= \int_{U} \prod_{i=1}^{N} \pi_i(u_i|\tau_i) \sum_{i=1}^{N} \log \frac{\pi_i(u_i|\tau_i)}{\pi_i^*(u_i|\tau_i)} du \\
&= \int_{U} \prod_{i=1}^{N} \pi_i(u_i|\tau_i) \log \frac{\pi_1(u_1|\tau_1)}{\pi_1^*(u_1|\tau_1)} du + \cdots + \int_{U} \prod_{i=1}^{N} \pi_i(u_i|\tau_i) \log \frac{\pi_N(u_N|\tau_N)}{\pi_N^*(u_N|\tau_N)} du \\
&= \sum_{i=1}^{N} \int_{u_i} \int_{u_{\text{-i}}} \pi_i(u_i|\tau_i) \pi_{\text{-i}}(u_{\text{-i}}|\tau_{\text{-i}}) \log \frac{\pi_i(u_i|\tau_i)}{\pi_i^*(u_i|\tau_i)} du_i du_{\text{-i}} \\
&= \sum_{i=1}^{N} \int_{u_i} \pi_i(u_i|\tau_i) \log \frac{\pi_i(u_i|\tau_i)}{\pi_i^*(u_i|\tau_i)} du_i \int_{u_{\text{-i}}} \pi_{\text{-i}}(u_{\text{-i}}|\tau_{\text{-i}}) du_{\text{-i}} \\
&= \sum_{i=1}^{N} D_{KL}(\pi_i(u_i|\tau_i) \parallel \pi_i^*(u_i|\tau_i)) \int_{u_{\text{-i}}} \pi_{\text{-i}}(u_{\text{-i}}|\tau_{\text{-i}}) du_{\text{-i}} \\
&= \sum_{i=1}^{N} D_{KL}(\pi_i(u_i|\tau_i) \parallel \pi_i^*(u_i|\tau_i)),
\end{aligned}
\tag{38}
$$

where $u_{-i}$ denotes the joint action without agent $i$, and $\pi_{-i}$ denotes the joint policy without agent $i$.

### C.4 TRANSFORM POLICY ITERATION

**Lemma 3 (Transform Policy Evaluation).** *Consider the transform Bellman backup operator $\mathcal{T}_{\pi_{jt}}^{tra}$ in Eq. (9) and a mapping $Q_{jt}^0 : \tau \times \widetilde{u} \to \mathbb{R}$ with $|\mathcal{A}| < \infty$, and define $Q_{jt}^{k+1} = \mathcal{T}_{\pi_{jt}}^{tra} Q_{jt}^k$. Then the sequence $Q_{jt}^k$ will converge to the soft joint Q-value of $\pi_{jt}$ as $k \to \infty$.*

*Proof.* Define the entropy augmented reward as

$$
r_\pi(\tau_t, \widetilde{u}_t) \triangleq r(\tau_t, \widetilde{u}_t) + \mathbb{E}_{\tau_{t+1} \sim p(\tau_{t+1}|\tau_t, \widetilde{u}_t)} [Q_{\text{jt}}(\tau_{t+1}, \widetilde{u}_{t+1})]
\tag{39}
$$

Therefore, the soft action value function is given by

$$Q_{\mathrm{jt}}(\tau_t, \widetilde{u}_t) \leftarrow r_{\pi_{\mathrm{jt}}}(\tau_t, \widetilde{u}_t) + \gamma \mathbb{E}_{\tau_{t+1} \sim p(\tau_{t+1}|\tau_t, \widetilde{u}_t), \widetilde{u}_{t+!} \sim \pi(\cdot|\tau_{t+1})} \left[ Q(\tau_{t+1}, \widetilde{u}_{t+1}) \right] \tag{40}$$

Apply the standard convergence results for policy evaluation Sutton & Barto (1999) and the assumption $|\mathcal{U}| < \infty$ is required to guarantee that the entropy augmented reward is bounded. $\square$

**Lemma 4 (Transform Policy Improvement).** *Let $\pi_{jt}^{old} \in \Pi$ and let $\pi_{jt}^{new}$ be the optimizer of the minimization problem defined in Eq. (11). Then $Q^{\pi_{jt}^{new}}(\tau_t, \widetilde{u}_t) \geq Q^{\pi_{jt}^{old}}(\tau_t, \widetilde{u}_t)$ for all $(\tau_t, \widetilde{u}_t) \in \tau \times \widetilde{\mathcal{U}}$ with $|\widetilde{\mathcal{U}}| < \infty$.*

*Proof.* Let $Q^{\pi_{\mathrm{jt}}^{\mathrm{old}}}$ and $V^{\pi_{\mathrm{jt}}^{\mathrm{old}}}$ be the corresponding soft joint Q-value and soft joint V-value under the old joint policy, and let $\pi_{\mathrm{jt}}^{\mathrm{new}}$ be defined as

$$
\begin{aligned}
\pi_{\mathrm{jt}}^{\mathrm{new}}(\cdot|\tau_t) &= \operatorname*{argmin}_{\pi \in \Pi} D_{\mathrm{KL}}(\pi(\cdot|\tau_t) \,\|\, \exp(Q^{\pi_{\mathrm{jt}}^{\mathrm{old}}}(\tau_t, \cdot) - \log Z^{\pi_{\mathrm{jt}}^{\mathrm{old}}}(\tau_t))) \\
&= \operatorname*{argmin}_{\pi \in \Pi} \mathbb{E}_{\widetilde{u}_t \sim \pi(\widetilde{u}_t|\tau_t)} \left[ \log \frac{\pi(\widetilde{u}_t|\tau_t)}{\exp(Q^{\pi_{\mathrm{jt}}^{\mathrm{old}}}(\tau_t, \widetilde{u}_t) - \log Z^{\pi_{\mathrm{jt}}^{\mathrm{old}}}(\tau_t))} \right] \\
&= \operatorname*{argmin}_{\pi \in \Pi} \mathbb{E}_{\widetilde{u}_t \sim \pi(\widetilde{u}_t|\tau_t)} \left[ \log \pi(\widetilde{u}_t|\tau_t) - Q^{\pi_{\mathrm{jt}}^{\mathrm{old}}}(\tau_t, \widetilde{u}_t) + \log Z^{\pi_{\mathrm{jt}}^{\mathrm{old}}}(\tau_t) \right]
\end{aligned} \tag{41}
$$

Let $J_{\pi_{\mathrm{jt}}^{\mathrm{old}}}(\pi(\cdot|\tau_t)) = \mathbb{E}_{\widetilde{u}_t \sim \pi(\widetilde{u}_t|\tau_t)} \left[ \log \pi(\widetilde{u}_t|\tau_t) - Q^{\pi_{\mathrm{jt}}^{\mathrm{old}}}(\tau_t, \widetilde{u}_t) + \log Z^{\pi_{\mathrm{jt}}^{\mathrm{old}}}(\tau_t) \right]$, then

$$\pi_{\mathrm{jt}}^{\mathrm{new}}(\cdot|\tau_t) = \operatorname*{argmax}_{\pi \in \Pi} J_{\pi_{\mathrm{jt}}^{\mathrm{old}}}(\pi(\cdot|\tau_t)) \tag{42}$$

From the definition of $\pi_{\mathrm{jt}}^{\mathrm{old}}$ and $\pi_{\mathrm{jt}}^{\mathrm{new}}$, we can show that $J_{\pi_{\mathrm{jt}}^{\mathrm{old}}}(\pi_{\mathrm{jt}}^{\mathrm{new}}(\cdot|\tau_t)) \leq J_{\pi_{\mathrm{jt}}^{\mathrm{old}}}(\pi_{\mathrm{jt}}^{\mathrm{old}}(\cdot|\tau_t))$. Hence,

$$\mathbb{E}_{\widetilde{\tau}_t \sim \pi_{\mathrm{jt}}^{\mathrm{new}}(\widetilde{u}_t|\tau_t)} \left[ \log \pi_{\mathrm{jt}}^{\mathrm{new}}(\widetilde{u}_t|\tau_t) - Q^{\pi_{\mathrm{jt}}^{\mathrm{old}}}(\tau_t, \widetilde{u}_t) \right] \leq \mathbb{E}_{\widetilde{u}_t \sim \pi_{\mathrm{jt}}^{\mathrm{old}}(\widetilde{u}_t|\tau_t)} \left[ \log \pi_{\mathrm{jt}}^{\mathrm{old}}(\widetilde{u}_t|\tau_t) - Q^{\pi_{\mathrm{jt}}^{\mathrm{old}}}(\tau_t, \widetilde{u}_t) \right] \tag{43}$$

Consider the relationship between soft V-function and soft Q-function, the Eq. (43) reduces to

$$V^{\pi_{\mathrm{jt}}^{\mathrm{old}}}(\tau_t) \leq \mathbb{E}_{\widetilde{u}_t \sim \pi_{\mathrm{jt}}^{\mathrm{new}}(\widetilde{u}_t|\tau_t)} \left[ Q^{\pi_{\mathrm{jt}}^{\mathrm{old}}}(\tau_t, \widetilde{u}_t) - \log \pi_{\mathrm{jt}}^{\mathrm{new}}(\widetilde{u}_t|\tau_t) \right] \tag{44}$$

Next, we consider the soft Bellman equation:

$$
\begin{aligned}
Q^{\pi_{\mathrm{jt}}^{\mathrm{old}}}(\tau_t, \widetilde{u}_t) &= r(\tau_t, \widetilde{u}_t) + \gamma \mathbb{E}_{\tau_{t+1} \sim p(\tau_{t+1}|\tau_t, \widetilde{u}_t)} \left[ V^{\pi_{\mathrm{jt}}^{\mathrm{old}}}(\tau_{t+1}) \right] \\
&\leq r(\tau_t, \widetilde{u}_t) + \gamma \mathbb{E}_{\tau_{t+1} \sim p(\tau_{t+1}|\tau_t, \widetilde{u}_t)} \left[ \mathbb{E}_{\widetilde{u}_{t+1} \sim \pi_{\mathrm{jt}}^{\mathrm{new}}(\widetilde{u}_{t+1}|\tau_{t+1})} \left[ Q^{\pi_{\mathrm{jt}}^{\mathrm{old}}}(\tau_{t+1}, \widetilde{u}_{t+1}) - \log \pi_{\mathrm{jt}}^{\mathrm{new}}(\widetilde{u}_{t+1}|\tau_{t+1}) \right] \right] \\
&\vdots \\
&\leq Q^{\pi_{\mathrm{jt}}^{\mathrm{new}}}(\tau_t, \widetilde{u}_t)
\end{aligned}
$$

$$\tag{45}$$

This completes the proof of Lemma 4 $\square$

Lemma 4 does not make the factorizable assumption on Q-value and thus is applicable to general settings. In other word, the derivation on the case of Q-value decomposition is a special case of Lemma 4. Here we provide a derivation example for monotonic linear value decomposition (MLVD): $Q_{\mathrm{jt}}(\tau, u) = \sum_{i=1}^{N} w_i(\tau_i) Q_i(\tau_i, u_i)$. Other value decomposition ways could have the similar derivation.

**Lemma 5 (Transform Policy Improvement with MLVD).** *If the soft joint Q-function satisfies $Q_{jt}(\tau, u) = \sum_{i=1}^{N} w_i(\tau_i) Q_i(\tau_i, u_i)$ and let $\pi_i^{new}$ be the optimizer of the minimization problem satisfied $\pi_i^{new} = \operatorname{argmin}_{\pi_i} D_{KL}(\pi_i(\cdot|\tau_i) \,\|\, \frac{\exp(w_i(\tau_i) Q_i^{\pi_i^{old}}(\tau_i, \cdot))}{Z^{\pi_i^{old}}(\tau_i)})$. Then $Q^{\pi_{jt}^{new}}(\tau_t, \widetilde{u}_t) \geq Q^{\pi_{jt}^{old}}(\tau_t, \widetilde{u}_t)$ for all $(\tau_t, \widetilde{u}_t) \in \tau \times \widetilde{\mathcal{U}}$ with $|\widetilde{\mathcal{U}}| < \infty$, where $\pi_{jt}^{new} = \prod_{i=1}^{N} \pi_i^{new}$ and $\pi_{jt}^{old} = \prod_{i=1}^{N} \pi_i^{old}$.*

*Proof.*

$$
\begin{aligned}
\pi_i^{new} &= \operatorname*{argmin}_{\pi_i} D_{\mathrm{KL}}(\pi_i(\cdot|\tau_i) \,\|\, \frac{\exp(w_i(\tau_i)Q_i^{\pi_i^{\mathrm{old}}}(\tau_i, \cdot))}{Z^{\pi_i^{\mathrm{old}}}(\tau_i)}) \\
&= \operatorname*{argmin}_{\pi_i} D_{\mathrm{KL}}(\pi_i(\cdot|\tau_i) \,\|\, \exp(w_i(\tau_i)Q_i^{\pi_i^{\mathrm{old}}}(\tau_i, \cdot) - \log Z^{\pi_i^{\mathrm{old}}}(\tau_i))) \\
&= \operatorname*{argmin}_{\pi_i} \mathbb{E}_{\widetilde{u}_i \sim \pi_i(\widetilde{u}_i|\tau_i)} \left[ \log \frac{\pi_i(\widetilde{u}_i|\tau_i)}{\exp(w_i(\tau_i)Q_i^{\pi_i^{\mathrm{old}}}(\tau_i, \widetilde{u}_i) - \log Z^{\pi_i^{\mathrm{old}}}(\tau_i)))} \right] \\
&= \operatorname*{argmin}_{\pi_i} \mathbb{E}_{\widetilde{u}_i \sim \pi_i(\widetilde{u}_i|\tau_i)} \left[ \log \pi_i(\widetilde{u}_i|\tau_i) - w_i(\tau_i)Q_i^{\pi_i^{\mathrm{old}}}(\tau_i, \widetilde{u}_i) + \log Z^{\pi_i^{\mathrm{old}}}(\tau_i) \right] \\
&= \operatorname*{argmax}_{\pi_i} \sum_{\widetilde{u}_i} \pi_i(\widetilde{u}_i|\tau_i) \left( w_i(\tau_i)Q_i^{\pi_i^{\mathrm{old}}}(\tau_i, \widetilde{u}_i) - \log \pi_i(\widetilde{u}_i|\tau_i) - \log Z^{\pi_i^{\mathrm{old}}}(\tau_i) \right)
\end{aligned}
\tag{46}
$$

According to the above equation, we can obtain $\sum_{\widetilde{u}_i} \pi_i^{\mathrm{new}}(\widetilde{u}_i|\tau_i) \left( w_i(\tau_i)Q_i^{\pi_i^{\mathrm{old}}}(\tau_i, \widetilde{u}_i) - \log \pi_i^{\mathrm{new}}(\widetilde{u}_i|\tau_i) \right) \geq$
$\sum_{\widetilde{u}_i} \pi_i^{\mathrm{old}}(\widetilde{u}_i|\tau_i) \left( w_i(\tau_i)Q_i^{\pi_i^{\mathrm{old}}}(\tau_i, \widetilde{u}_i) - \log \pi_i^{\mathrm{old}}(\widetilde{u}_i|\tau_i) \right)$.

Consider the relationship between soft V-function and soft Q-function, we can show that

$$
\begin{aligned}
\mathbb{E}_{\widetilde{u} \sim \pi_{\mathrm{jt}}^{\mathrm{new}}(\widetilde{u}|\tau)} &\left[ Q^{\pi_{\mathrm{jt}}^{\mathrm{old}}}(\tau, \widetilde{u}) - \log \pi_{jt}^{\mathrm{new}}(\widetilde{u}|\tau) \right] \\
&= \sum_{\widetilde{u}} \pi_{\mathrm{jt}}^{\mathrm{new}}(\widetilde{u}|\tau) \sum_i \left( w_i(\tau_i)Q_i^{\pi_i^{\mathrm{old}}}(\tau_i, \widetilde{u}_i) - \log \pi_i^{\mathrm{new}}(\widetilde{u}_i|\tau_i) \right) \\
&= \sum_i \sum_{\widetilde{u}_{\text{-}i}} \pi_{\text{-}i}^{\mathrm{new}}(\widetilde{u}_{\text{-}i}|\tau_{\text{-}i}) \sum_{\widetilde{u}_i} \pi_i^{\mathrm{new}}(\widetilde{u}_i|\tau_i) \left( w_i(\tau_i)Q_i^{\pi_i^{\mathrm{old}}}(\tau_i, \widetilde{u}_i) - \log \pi_i^{\mathrm{new}}(\widetilde{u}_i|\tau_i) \right) \\
&\geq \sum_i \sum_{\widetilde{u}_{\text{-}i}} \pi_{\text{-}i}^{\mathrm{old}}(\widetilde{u}_{\text{-}i}|\tau_{\text{-}i}) \sum_{\widetilde{u}_i} \pi_i^{\mathrm{old}}(\widetilde{u}_i|\tau_i) \left( w_i(\tau_i)Q_i^{\pi_i^{\mathrm{old}}}(\tau_i, \widetilde{u}_i) - \log \pi_i^{\mathrm{old}}(\widetilde{u}_i|\tau_i) \right) \\
&= V^{\pi_{\mathrm{jt}}^{\mathrm{old}}}(\tau_t)
\end{aligned}
\tag{47}
$$

Next, we consider the soft Bellman equation:

$$
\begin{aligned}
Q^{\pi_{\mathrm{jt}}^{\mathrm{old}}}(\tau, \widetilde{u}) &= r(\tau, \widetilde{u}) + \gamma \mathbb{E}_{\tau'} \left[ V^{\pi_{\mathrm{jt}}^{\mathrm{old}}}(\tau') \right] \\
&\leq r(\tau, \widetilde{u}) + \gamma \mathbb{E}_{\tau'} \left[ \mathbb{E}_{\widetilde{u}' \sim \pi_{\mathrm{jt}}^{\mathrm{new}}} \left[ Q^{\pi_{\mathrm{jt}}^{\mathrm{old}}}(\tau_{t+1}, \widetilde{u}_{t+1}) - \log \pi_{\mathrm{jt}}^{\mathrm{new}}(\widetilde{u}_{t+1}|\tau_{t+1}) \right] \right] \\
&\vdots \\
&\leq Q^{\pi_{\mathrm{jt}}^{\mathrm{new}}}(\tau_t, \widetilde{u}_t)
\end{aligned}
\tag{48}
$$

This completes the proof of Lemma 5 $\qquad\square$

**Theorem 6** (**Transform Policy Iteration**). *Repeated application of Transform Policy Evaluation and Transform Policy Improvement, we can obtain a sequence $Q^k$ and this sequence will converge towards the optimal soft Q-function $Q_{soft}^*$, while the corresponding sequence of policies will converge towards the optimal policy $\pi_{jt}^*$.*

*Proof.* Let $\pi_{\mathrm{jt}}^i$ be the policy at iteration $i$. By Lemma 4, we can know that the sequence $Q_{\mathrm{soft}}^{\pi_{\mathrm{jt}}^i}$ is monotonically increasing. Since $Q_{\mathrm{soft}}^{\pi_{\mathrm{jt}}}$ is bounded above for $\pi_{\mathrm{jt}} \in \Pi$(both the reward and entropy are bounded), the sequence converges to some $\pi_{\mathrm{jt}}^*$. We will still need to show that $\pi_{\mathrm{jt}}^*$ is indeed optimal. At convergence, it must be case that $J_{\pi_{\mathrm{jt}}^*}(\pi_{\mathrm{jt}}^*(\cdot|\tau_t)) \leq J_{\pi_{\mathrm{jt}}^*}(\pi_{\mathrm{jt}}(\cdot|\tau_t))$ for all $\pi_{\mathrm{jt}} \in \Pi$, $\pi_{\mathrm{jt}} \neq \pi_{\mathrm{jt}}^*$. Using the same iterative argument as in the proof of Lemma 4, we get $Q^{\pi_{\mathrm{jt}}^*}(\tau_t, \widetilde{u}_t) > Q^{\pi_{\mathrm{jt}}}(\tau_t, \widetilde{u}_t)$ for all $(\tau_t, \widetilde{u}_t) \in \tau \times \widetilde{\mathcal{U}}$. Therefore, the soft action-value of any other policy in $\Pi$ is lower than that of the converged policy. Hence, $\pi_{\mathrm{jt}}^*$ is optimal in $\Pi$. $\qquad\square$

## D   THE ALGORITHM OF IGNT-MAC

Algorithm 1 introduces the complete training procedure of IGNT-MAC.

---
**Algorithm 1** IGNT-MAC
---
1: **for** episode = 1 to max-training-episode **do**
2:   Initialize the environment and obtain the initial state and observations;
3:   **for** $t = 1$ to max-episode-length **do**
4:     **for** each agent $\{a_i\}_{i=1}^N$ **do**
5:       Select a action from the individual policy $u_i \sim \pi_i(\cdot|\tau_i)$;
6:     **end for**
7:     Transform the joint action by an normalized transformation $\widetilde{u} = F(u)$
8:     Execute the transformed actions $\widetilde{u} = (\widetilde{u}_1, \cdots, \widetilde{u}_N)$ to obtain shared reward $r$ and the next observation $o_i$;
9:     Store $(o, u, r, o^{\text{next}}, done)$ in replay buffer D;
10:   **end for**
11:   Sample a random minibatch of $M$ samples from D:$(o_m, u_m, r_m, o_m^{\text{next}}, done_m)$;
12:   **for** agent $i = 1$ to $N$ **do**
13:     Update individual policy $\pi_i$: $\phi_i \leftarrow \phi_i + \beta_\phi \nabla_{\phi_i} \mathcal{L}_{\pi_i}$ ;
14:   **end for**
15:   Update the critic: $\theta \leftarrow \theta - \beta_\theta \nabla_\theta \mathcal{L}_Q$ ;
16:   Update the affine transformation: $\psi \leftarrow \psi - \beta_\psi \nabla_\psi \mathcal{L}_\pi$ ;
17:   **if** step%$C$ == 0 **then**
18:     Update target individual policy: $\overline{\phi}_i = (1 - \alpha)\overline{\phi}_i + \alpha\phi_i$;
19:     Update target critic: $\overline{\theta}_i = (1 - \alpha)\overline{\theta}_i + \alpha\theta_i$;
20:   **end if**
21: **end for**
---

## E   DETAILS OF ENVIRONMENTS

### E.1   THE STARCRAFT MULTI-AGENT CHALLENGE

The StarCraft Multi-Agent Challenge (SMAC) Samvelyan et al. (2019) environment is an popular platform for researching MARL, which is inspired by the real-time strategy (RTS) game StarCraft II. The SMAC environment offers a competitive setting for agents, where they are required to collaborate or engage in competition on a virtual StarCraft II game map. Specifically, the environment comprises two teams: the Red team and the Blue team. The Red team is controlled by a MARL method, where each agent's decision-making relies on its own action-observation historical trajectory. On the other hand, the Blue team (enemy) is controlled by the built-in game AI that utilizes handcrafted heuristics. Furthermore, SMAC encompasses a variety of StarCraft II micromanagement scenarios typically categorized into three level of difficulty: *Easy, Hard*, and *Super-Hard*. In our experiments, we consider six scenarios including 1c_3s_5z, 8m, 2c_vs_64zg, 3s_vs_5z, 25m and MMM2, to evaluate the effectiveness of the IGTO framework. The exhaustive list of challenges is presented in Table 1. Figure 4 shows the visualization of all scenarios utilized in our experiments.

Table 1: The details of StarCraft II challenges [SMAC Samvelyan et al. (2019)].

| Map Name | Ally Units | Enemy Units |
|---|---|---|
| 1c3s5z | 1 Colossus, 3 Stalkers & 5 Zealots | 1 Colossus, 3 Stalkers & 5 Zealots |
| 8m | 8 Marines | 8 Marines |
| 2c_vs_64zg | 2 Colossi | 64 Zerglings |
| 3s_vs_5z | 3 Stalkers | 5 Zealots |
| 25m | 25 Marines | 25 Marines |
| MMM2 | 1 Medivac, 2 Marauders & 7 Marines | 1 Medivac, 3 Marauders & 8 Marines |

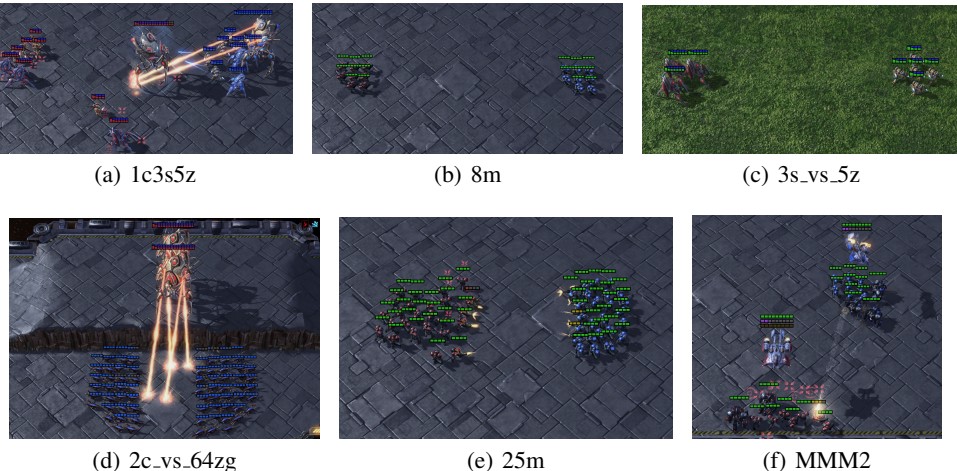

Figure 4: The screenshots of SMAC scenarios utilized in our experiments.

During each time step, agents get local observations, which are composed of agent movement, enemy, ally and agent unit features, within their sight. The dimensionality of observation vector may vary depend on the specific environment configuration and the types of units existing within the scenario. Specifically, the feature vector of observation includes attributes such as health, unit_type, shield, relative x, relative y and distance for both allied and enemy units. The features of agent unit contain its shield, health and unit_type. Furthermore, the action space of each agent comprises four features: move direction, no-option, stop, and attack target. Deceased agents are restricted to selecting the no-option feature, while living agents are unable to choose it. Each agent can choose to either stop or move in one of the four cardinal directions: north, east, south or west. Besides, the agent is permitted to execute the attack action only if the enemy is within the shooting range or field of attack. The detailed feature of observation and action for each agent are shown in Table 2.

Table 2: The feature size of observation and action for scenarios of SMAC.

| Map Name | Category | Move | Enemy | Ally | Own | Attack_id |
|----------|----------|------|-------|------|-----|-----------|
| 1c3s5z | *Easy* | 4 | (9, 9) | (8, 9) | 5 | 9 |
| 8m | | 4 | (8, 5) | (7, 5) | 1 | 8 |
| 2c_vs_64zg | *Hard* | 4 | (64, 5) | (1, 6) | 2 | 64 |
| 3s_vs_5z | | 4 | (5, 6) | (2, 6) | 2 | 5 |
| 25m | | 4 | (25, 5) | (24, 5) | 1 | 25 |
| MMM2 | *Super-Hard* | 4 | (12, 8) | (9, 8) | 4 | 12 |

In these combat scenarios, the goal is to maximize the win rate and episode reward. Note that all agents obtain the shared global reward, which corresponds to the total damage inflicted on all enemy agents combined. The reward scheme is that agents earns 10 points for successfully eliminating an enemy unit. Moreover, when all enemies are collectively eliminated, each agent is granted a bonus of 200 points. The cumulative reward is normalized to a range of 20.

## E.2 THE MULTI-AGENT PARTICLE ENVIRONMENT

The Multi-agent Particle Environment(MPE) provides a configurable two-dimensional grid world populated with a group of autonomous agents, where each agent is capable of perceiving the environment and taking actions to achieve its specific objective. To perform our experiments, we adopt some cooperative games in MPE including Cooperative Navigation, Modified Predator-prey, Physical Deception 2 and Physical Deception 4, as shown in Figure 5, and the detailed introduction and settings of these scenarios are provided as bellow.

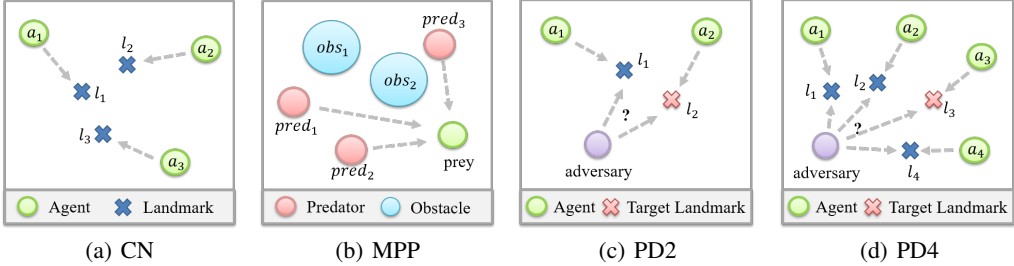

Figure 5: The visualization of the scenarios in MPE, including Cooperative Navigation(CN), Modified Predator-prey(MPP), Physical Deception 2(PD2) and Physical Deception 4(PD4).

**Cooperative Navigation** In this scenario, the objective is for three agents to collaborate by taking physical actions in order to reach a designated set of three landmarks. The agents receive observations regarding the relative positions of other agents and landmarks, and their collective reward is based on the proximity of any agent to each landmark. In essence, the agents must cover all of the landmarks successfully. Additionally, as the agents occupy physical space, they receive penalties for colliding with each other. The agents learn to deduce the specific landmark they need to reach, navigate towards it, and at the same time, avoid collisions with other agents.

**Modified Predator-prey** In this modified version of the traditional predator-prey game, three cooperating agents with slower speeds are tasked with pursuing a faster adversary within a randomly generated environment. The environment is further obstructed by two large landmarks. Whenever the cooperative agents successfully collide with the adversary, they receive rewards, while the adversary incurs penalties. The agents have access to observations regarding the relative positions and velocities of other agents, as well as the positions of the landmarks.

**Physical Deception 2** In this scenario, a pair of agents work together in a cooperative manner to reach a specific target landmark from a total of two available landmarks. The agents receive rewards based on the minimum distance between any agent and the target landmark, indicating that only one agent needs to reach the target. However, there is also a lone adversary in pursuit of the target landmark. The twist is that the adversary is unaware of which landmark is the correct one. Consequently, the cooperating agents, who face penalties based on the distance between the adversary and the target, learn to disperse and cover all the landmarks to deceive the adversary effectively.

**Physical Deception 4** In this scenario, a group of four agents collaborates to reach a specific target landmark from a set of four available landmarks. Their rewards are determined by the minimum distance between any agent and the target landmark, indicating that only one agent needs to successfully reach it. However, there is a lone adversary that also aims to reach the target landmark. The twist lies in the fact that the adversary lacks knowledge of which landmark is the correct one. As a result, the cooperative agents, who face penalties based on the distance between the adversary and the target, learn to disperse and cover all the landmarks in order to deceive and confuse the adversary.

# F DETAILS OF IMPLEMENTATION

## F.1 ADJUSTMENTS OF THE BASELINES

The adjustments of the baselines including VDN, QMIX, QTRAN and FOP are listed as follows: VDN, QMIX, and QTRAN are value-based decomposition MARL methods. Consequently, we incorporate individual policy network and target policy network for each agent. The original individual Q-value functions are retained to facilitate the learning of critic in IGNT-MAC variants. FOP, on the other hand, is a multi-agent actor and critic method that combines value decomposition with entropy regularization. Hence, we maintain the original architecture of FOP.

## F.2 Hyperparameters

All the individual policy networks consist of identical components, including two linear layers and one GRUCell layer with ReLU activation. The number of hidden units in each layer is set to 64. The architecture of individual Q-value function networks, which is shared across all value-based methods, comprises a Gated Recurrent Unit (GRU) combined with a fully-connected layer before and after. The learning rate for critic is $10^{-3}$ and the learning rate for actor is $10^{-4}$. Furthermore, $\gamma$ is set to 0.99. The replay buffer contains a collection of 5000 episodes. During the training process, batches with 32 episodes are sampled uniformly from the replay buffer. The training is conducted on fully unrolled episodes. After each episode, a single gradient descent step is performed to update the parameters of networks. The setting of all hyperparameters for the training are presented in Table 3.

Table 3: Hyperparameters for training.

| name | Value | Description |
|---|---|---|
| difficulty | 7 | the difficulty of the game |
| n_steps | $2 \times 10^6$ | Maximum steps until the end of training |
| buffer_size | 5000 | capacity of replay buffer |
| batch_size | 32 | number of samples from each update |
| evaluate_cycle | 5000 | how often to evaluate the model |
| $lr_{\pi_i}$ | $5 \times 10^{-4}$ | learning rate for individual policies |
| $lr_Q$ | $5 \times 10^{-3}$ | learning rate for critic |
| $C$ | 400 | how often target networks update |
| $\gamma$ | 0.99 | discount factor |
| $\alpha$ | 0.999 | the parameter used for soft updating target networks |

## G   Additional Experiments Results

Figure 6 and Figure 7 show the learning curves of COMA, MAAC, QMIX and DOP groups in terms of average success rates and episode rewards in six SMAC scenarios including 1c3s5z, 8m, 2c_vs_64zg, 25m and MMM2.

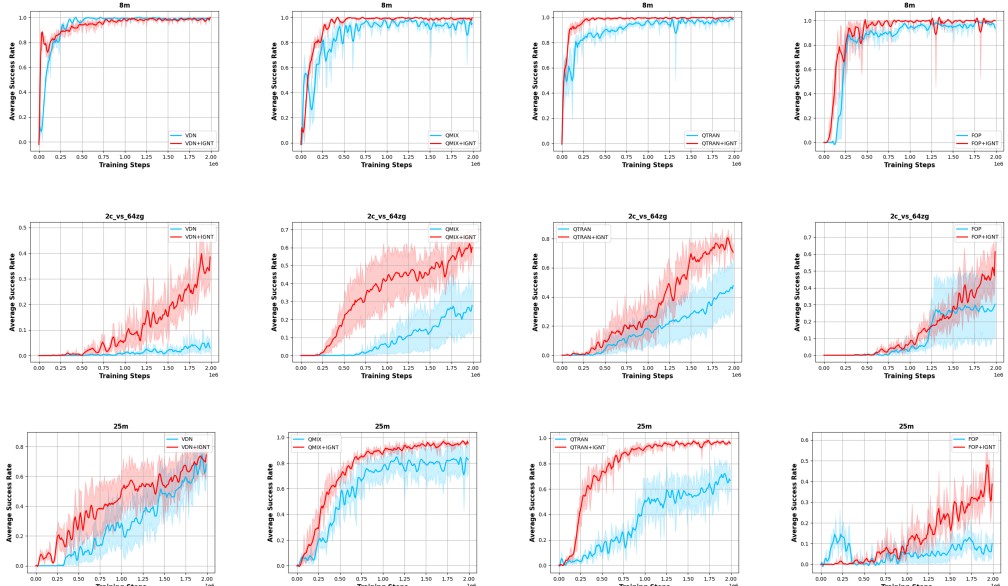

Figure 6: The performance in terms of average success rates for some baselines (VDN, QMIX, QTRAN and FOP) and their variants under the framework of IGNT-MAC on other scenarios of StarCraft II benchmark including 8m, 2c_vs_64zg and 25m. In particular, each row exhibits the results of different groups in the same scenario.

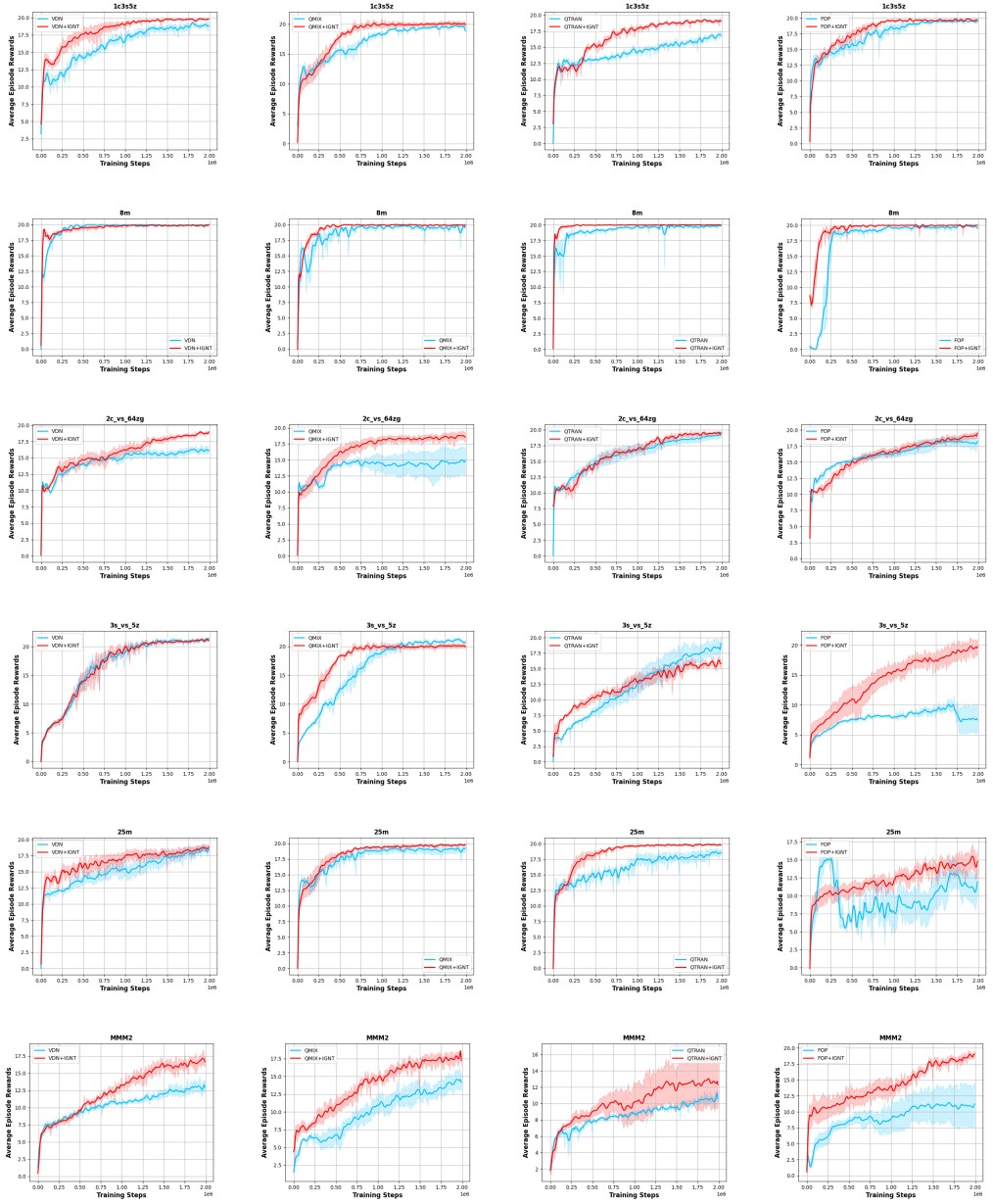

Figure 7: The performance in terms of average episode rewards for baselines (VDN, QMIX, QTRAN and FOP) and their variants under the framework of IGNT-MAC on various scenarios of StarCraft II benchmark including 1c3s5z, 8m, 2c_vs_64zg, 3s_vs_5z, 25m and MMM2. In particular, each row exhibits the results of different groups in the same scenario.