# OpenReview forum: "IGTO: Individual Global Transform Optimization for Multi-Agent Reinforcement Learning"
_ICLR.cc/2024/Conference — Submitted to ICLR 2024_

### Official Review · Reviewer_Kp6o · 2023-10-26

**Soundness:** 4 excellent
**Presentation:** 4 excellent
**Contribution:** 4 excellent
**Rating:** 8
**Confidence:** 4

**Summary:**

This paper introduces the Individual-Global Transform-Optimal (IGTO) condition, allowing for inconsistent individual-global actions while ensuring equivalent policy distributions. The authors also develop the Individual-Global Normalized Transformation (IGNT) rule, which can be integrated into existing CTDE-based algorithms and helps achieve optimum convergence of individual-global policies. Extensive experiments demonstrate the effectiveness of the proposed method.

**Strengths:**

1.	This paper proposes the IGTO condition, which alleviates the limitations of the IGM condition.
2.	The paper presents a novel method for obtaining individual transformed policy improvement through global joint policy optimization in the centralized training procedure.
3.	The methods are designed with theoretical guarantees, and the experiments provide evidence of their effectiveness.

**Weaknesses:**

Overall, this is a good paper and I have not found significant weakness yet.
I would lean towards accepting this paper. Meanwhile, I will also pay attention to the opinions and discussions of other reviewers.

There are some typos such as Eq.14.
Some symbols in the definitions lack explanations, making it difficult to understand such as Eq.5.

**Questions:**

1.	IGTO is based on Theorem 1 where the transformation is performed sequentially. Whether the order will influence the final results?
2.	From Eq.13, the joint policy is restricted to some set of intractable policies. Whether this global operation may lead to the suboptimal results of individual policies.
3.	How the method will perform in more complex scenarios in SC2 such as corridor, 6h_vs_8z?

---

> ### Author Response · Authors · 2023-11-22
> **Initial Response to Reviewer Kp6o**
>
> # Thank you for recognizing our innovation and promising results. Below, we make the responses to your concerns.
>
> **Question 1: IGTO is based on Theorem 1 where the transformation is performed sequentially. Whether the order will influence the final results?**
>
> Thanks for this comment. **Theorem 1** shows that, if the transformation is performed sequentially and the Jacobian determinant of transformation satisfies $\lvert \textbf{G}_i \rvert = 1$, the IGTO condition will be guaranteed. In fact, the order of performing transformation may be changed. For convenience, we take three agents as example, and change the sequence from the original setting $(f_1,f_2,f_3)$ to a new setting $(f_2,f_3,f_1)$. We summary the conclusion, and its proof is deferred to **Appendix C.1** (Supplementary material):
>
> **Remark** *Considering a factorizable MARL task with three agents, i.e., $N=3$. We first perform transformation $f_2$, then transformation $f_3$, finally transformation $f_1$. If the Jacobian determinant of transformation satisfies $\lvert \textbf{G}_i \rvert = 1, i = 1, 2, 3$, then the IGTO condition is provable.*
>
> The above conclusion can also be extended to general factorizable MARL tasks. In other words, the order of performing transformations will not influence the final results.
>
> **Question 2: From Eq.(13), the joint policy is restricted to some set of intractable policies. Whether this global operation may lead to the suboptimal results of individual policies.**
>
> Sorry for making you confused on this sentence, which actually results from our writing mistake.
> In the original manuscript, this sentence is: ``Therefore, we will restrict the joint policy to some set of intractable policies ...". The word "intractable" should be replaced with "tractable".
>
> In the tractable policies, this global operation would not lead to the suboptimal results of individual policies, according to our Lemma 2 in the manuscript.
>
> We have revised the sentence in the updated version. Thanks for the comment again.
>
> **Question 3: How the method will perform in more complex scenarios in SC2 such as corridor, 6h\_vs\_8z?**
>
> Thanks for your suggestion. We conduct additional experiments on corridor and 6h\_vs\_8z scenarios, which are categorized as Super-Hard, to compare these variants within our framework(IGNT-MAC) and the baselines methods. For the case that both the baseline and IGNT-MAC can hardly win on these scenarios (the analysis is given the next paragraph), we use average episode rewards to show the difference. We evaluate each method four times with different seeds and report the mean and standard deviation. We show the results in the following table:
>
> Table 1. The average episode rewards for the baselines and their variants under our framework.
> |      | QTRAN | QTRAN+IGNT(ours)   | FOP | FOP+IGNT(ours)     |
> | :---        |    :----:   |      :---: |       :----:   |      :---: |
> | 6h\_vs\_8z      | 11.274±0.239      | **12.805±0.273**   |  8.429±1.187      | **10.782±0.319**   |
> | corridor   |  7.853±0.396        | **8.602±0.236**      |   5.176±1.239      | **9.214±0.428**   |
>
> The experimental results shows that these integrated with IGNT rule achieve better performance compared to the original baselines in these Super-Hard scenarios. The variants under the framework of IGNT-MAC as well as all baselines fail to win in these Super-Hard scenarios. The reason is that, all approaches employ noise-based exploration, the agents face challenges in identifying states that are worth exploring and struggle to effectively coordinate their exploration efforts towards those states. For example, in 6h vs 8z scenario including 6 Hydralisks and 8 Zealots, the only winning strategy is requiring all Hyralisks to ambush enemy units in their path and then attack together when the Zealots approaching (**Liu et al. (2021)**). Thus, it is extremely challenging for any methods to pick up this strategy without improved exploration technique. This demonstrates that efficient exploration for MARL is still a challenging problem.
>
> Reference
>
> [1] Iou-Jen Liu, Unnat Jain, Raymond A Yeh, and Alexander Schwing. Cooperative exploration for multi-agent deep reinforcement learning. In International Conference on Machine Learning, pp. 6826–6836. PMLR, 2021
>
> **Question 4: There are some typos such as Eq.(14). Some symbols in the definitions lack explanations, making it difficult to understand such as Eq.(5).**
>
> Thanks for your careful review. We have clarified a few mathematical symbols, revised these typos and polished this manuscript carefully, according to your suggestions.
>
> **Please note that the updated manuscript for the revision has been uploaded.**

---

### Official Review · Reviewer_gq26 · 2023-11-01

**Soundness:** 2 fair
**Presentation:** 2 fair
**Contribution:** 2 fair
**Rating:** 5
**Confidence:** 3

**Summary:**

The paper introduces a new condition called Individual-Global Transform-Optimal (IGTO) to allow inconsistent individual-global actions while ensuring the equivalency of their policy distributions in Multi-Agent Reinforcement Learning (MARL). The authors propose a rule called Individual-Global Normalized Transformation (IGNT) to satisfy the IGTO constraint and integrate it into existing MARL algorithms. Theoretical proofs show that individual-global policies can converge to the optimum under the IGNT rule. The authors demonstrate the proposed method effectiveness through experiments on StarCraft Multi-Agent Challenge (SMAC) and Multi-Agent Particle Environment (MPE).

**Strengths:**

- Proposes the Individual-Global Normalized Transformation (IGNT) rule, which can be seamlessly integrated into existing MARL algorithms, offering a practical solution to satisfy the IGTO condition

- Providing detailed explanations and proofs in the appendix, enhancing the transparency and reproducibility of the proposed approach

**Weaknesses:**

- This work is less novelty, by adding the normalized transformation to FOP. The majority of the proofs are similar to FOP.
- IGNT is more suitable for policy-based methods. However, chosen baselines are almost value-based methods. Especially, MADDPG in MPE is ignored.
- Lack of motivation about why we need the normalized transformation, especially if the transformation is invertible.
- Lack of ablation about the chosen transformation function. For example, an Identity matrix, which $|G_i| = 1$ can be chosen, so that after transformation, actions are the same.

**Questions:**

- Could you provide an example of the normalized transformation? Especially, since the action is discrete in SMAC, u_i is an index of one action.
- It is unclear to me about how to adapt IGNT to value-based decomposition MARL methods, e.g., QMIX. If incorporating individual policy networks and target policy networks for each agent into QMIX, the ablation study should include such modifications.
- Why FOP cannot learn well in 3s_vs_5z? FOP can reach around 70% win-rate in their original paper in MMM2.

**Details Of Ethics Concerns:**

No ethics concern.

---

> ### Author Response · Authors · 2023-11-22
> **Initial Response to Reviewer gq26 (Part 1/2)**
>
> # Thank you for careful review and some valuable comments, below we make the responses to your concerns:
>
> **Q1 (w.r.t Weakness 1): This work is less novelty, by adding the normalized transformation to FOP. The majority of the proofs are similar to FOP.**
>
> We make the clarification as below:
>
> (i) Novelty: a) propose a new condition called IGTO with well theoretical guarantee, which relaxes the constraint of rigorous equivalency of individual-global actions; b) design an individual-global normalized transformation (IGNT) to satisfy this IGTO condition, which has also theoretical guarantee of optimal individual-global policies. *At the same time, please note that, the proposed normalized transformation (i.e., IGNT) can implanted into many existing CTDE-based algorithms, including not only FOP you argued, but also QMIX, QTRAN, VDN, etc.*
>
> (ii) Different FOP with the IGO constraint, ours is with the new proposed IGTO condition. Hence, the corresponding proof process has still the essential differences especially in the aspect of extra transformation.
>
> **Q2 (w.r.t Weakness 2): IGNT is more suitable for policy-based methods. However, chosen baselines are almost value-based methods. Especially, MADDPG in MPE is ignored.**
>
> According to your suggestion, we conduct the experiment by using MADDPG as the baseline in the MPE benchmark. The scenarios of MPE consist of Cooperative Navigation(CN), Modified Predator-prey(MPP), Physical Deception 2(PD2) and Physical Deception 4(PD4). The results are reported in the following Table. The experimental results show that the variant within our IGNT-MAC framework consistenly achieve better performance than the baseline MADDPG.
>
> Table 1. The average episode rewards for the MADDPG and its variant under our framework.
> |      | CN | MPP   | PD2 | PD4     |
> | :---        |    :----:   |      :---: |       :----:   |      :---: |
> | MADDPG      | -216.57±10.36      | 76.29±4.92   |  18.30±5.07      | 21.82±2.63   |
> | **MADDPG+IGNT(ours)**   |  **-184.38±6.23**        | **84.25±3.26**      |   **30.63±4.52**     | **35.39±3.81**   |
>
> **Q3 (w.r.t Weakness 3): Lack of motivation about why we need the normalized transformation, especially if the transformation is invertible.**
>
> For the CTDE paradigm, during training, the joint optimal actions of multi-agent are constrained to equal to individual-agent optimal actions, named the IGM condition. However, in real environments, the individual-global action equivalency would not be satisfied if the agent cannot obtain global information (e.g., other agents' situations). Hereby, we seek to relax the complete equivalency of individual-global actions. To this end, in favor of the consistency of policy distribution, we attempt to introduce the normalized transformation constraint between individual and global actions. The requirement of transformation invertibility is that we expect to have a well matching relation between individual actions and global actions. Both the theory and the experiments verify the effectiveness of the transformation.

---

> ### Author Response · Authors · 2023-11-22
> **Initial Response to Reviewer gq26 (Part 2/2)**
>
> **Q4 (w.r.t Question 1): Could you provide an example of the normalized transformation? Especially, since the action is discrete in SMAC, $u_i$ is an index of one action.**
>
> The question you concerned should be how to address the discrete actions for environment interaction.  We do clarify that, in implementation, the transformation function is preformed in the continuous numerical space, i.e., its inputs and outputs are probability values. Like plenty of single/multi-agent RL methods, the continuous probability values can be sampled to discrete actions. Hence, the example of the normalized transformation need not be discussed.
>
> **Q5 (w.r.t Question 2): It is unclear how to adapt IGNT to value-based decomposition MARL methods, e.g., QMIX.**
>
> Here we give the description of adapt our IGNT to the value-based decomposition method, such as QMIX.
>
> a) At the beginning of training, we initialize the environment and obtain the initial state and observations. b) Then, a action is selected from the individual policy according to the action-observation historical trajectory for each agent and the joint action, together formed by all agents, is transformed by the normalized transformation. c) Next, the transformed action is executed to obtain shared reward and the next observation. When the episode is terminated, the information of episode will be stored in replay buffer. d) After that, a batch of samples are sampled randomly from replay buffer and update each individual policy, the critic and the normalized transformation. e) After training every 200 steps, the individual target policy and critic will be updated.
>
> Note that these value-based composition methods need minor modifications to fit our IGNT-MAC framework. Concretely, for value-based composition method, we incorporate individual policy network and target network for each agent, and the original individual Q-value functions are retained to facilitate the learning of critic in the IGNT-MAC variant.
>
> **Q6 (w.r.t Question 3): Why FOP cannot learn well in 3s\_vs\_5z? FOP can reach around 70\% win-rate in their original paper in MMM2.**
>
> The reason why FOP cannot learn well in 3s\_vs\_5z is two-fold:
>
> i) Scenario characteristic: the scenario 3s\_vs\_5z, consisting of 3 allied stalkers and 5 enemy zealots, is an asymmetric task that requires precise control such as keeping a distance, evading attacks from the enemy units to win consistently. In particular, the enemy zealots have more powerful attacks and sturdy armor compared to allied stalkers. Therefore, the allied agents have to leverage agents' advantages and adopt flexible tactics to win in battle. To better understand the learned policy by FOP in 3s\_vs\_5z, we examine the learned behavior of each agent from the battle replay. We observe that, FOP tend to learn a specific strategy that, agents initially move right, and then engage enemies once they are in the shooting range, without making full use of fighting states of multi-agents themself vs enemies.
>
> ii) FOP characteristic: FOP is a Maximum-entropy MARL method that learns an optimal joint policy for entropy-regularized Dec-POMDP rather than the original MDP. As pointed in DMAC~\cite{su2022divergence}, the converged policy in FOP may be biased, which is one possible reason that lead to unsatisfied performance in some complicated scenarios such as 3s\_vs\_5z.
>
> For the question that FOP can reach around 70\% win-rate in their original paper in MMM2, we make the clarification that all comparison results of FOP in SMAC benchmark are carefully produced from the official code (https://github.com/liyheng/FOP).
>
> **Please note that the updated manuscript for the revision has been uploaded.**

---

> > ### Comment · Reviewer_gq26 · 2023-11-22
> >
> > Actually, I have the same concern as Reviewer ujuj pointed out about the discrete actions. The Theorem 1 will not hold. Also, if it holds, an Identity matrix is one type of transformation that satisfies Theorem 1, individual global transform optimality exists without any changes in actions.
> >
> > That is also why I have Q4 and Q5. Maybe the authors can use an example to illustrate. Such as how to transform the prediction values 0.85,0.05, 0.1 for A, B, C to new actions and perform normalized transformation, etc.

---

> > > ### Author Response · Authors · 2023-11-22
> > > **Response to Reviewer gq26**
> > >
> > > Thanks for your timely reply.
> > >
> > > For the same concern, please see the reply to Reviewer ujuj.
> > >
> > > For the question how to transform the prediction values, we use a three-layer neural network
> > > following by softmax for the probability value. The network is learnable during training.
> > >
> > > We hope you can understand the answer. Thanks again.

---

### Official Review · Reviewer_s6zd · 2023-11-02

**Soundness:** 2 fair
**Presentation:** 3 good
**Contribution:** 2 fair
**Rating:** 6
**Confidence:** 4

**Summary:**

This paper presents an individual-global action-transformed condition named IGTO, which releases the IGM or IGO restriction. To achieve this, it designs a bijective function for each agent to convert actions and proves that the converted actions can converge to the optimal policy. This method can be seamlessly implanted into many CTDE-based algorithms.

**Strengths:**

1. This paper attempts to find a solution to the optimal action of multi-agent from a new perspective, which is interesting.

2. IGNT is suitable for discrete and continuous actions and can be used in many agent reinforcement learning methods.

**Weaknesses:**

1. The motivation of this work is not clearly presented. Why do we need transformed actions? What are the benefits brought by this new method?

2. The experiments of this method are not convincing. More IGM methods such as QPLEX [ICLR 21], WQMIX [NeurIPS 21], ResQ [NeurIPS 22], etc. could be added to prove the effect of this work.

3. This paper provides a proof of improvement for the policy-based method, however, the proof of the value decomposition method is not yet clear.

**Questions:**

1.  IGM or IGO strictly requires the optimal joint actions to be consistent with the optimal individual behaviors, which may lead to unsatisfied performance in some complicated environments. Could you describe a scenario where the IGM or IGO conditions could lead to poor performance?

2. A closely related work is missing. ResQ [1] is a decomposition-based MARL method which learn a value decomposition by using a nonlinear function.

3. For the discrete or continuous action space of an agent, how to ensure that all actions converted by the bijective function f_i are valid, and they should include all actions in the raw action space. It is unclear to me how it ensures this condition.

4. In value-based decomposition MARL methods，Do $Q_{jt}^* (\tilde{u} |\tau)$ and $[Q_i^* ( \tilde{u}_i|\tau)]$ satisfy IGM condition ? Can the author provide the complete training procedure based on value decomposition such as QMIX.

References

[1] ResQ: A Residual Q Function-based Approach for Multi-Agent Reinforcement Learning Value Factorization, NeurIPS 2022

---

> ### Author Response · Authors · 2023-11-22
> **Initial Response to Reviewer s6zd (Part 1/2)**
>
> # Thank you for recognizing our innovation and promising results. Below, we make the responses to your concerns:
>
> **Q1 (w.r.t Weakness 1): The motivation of this work is not clearly presented. Why do we need transformed actions? What are the benefits brought by this new method?**
>
> For the CTDE paradigm, during training, the joint optimal actions of multi-agent are constrained to equal to individual-agent optimal actions, named the IGM condition. However, in real environments, the individual-global action equivalency would not be satisfied if the agent cannot obtain global information (e.g., other agents' situations). Hereby, we seek to relax the complete equivalency of individual-global actions. To this end, in favor of the consistency of policy distribution, we attempt to introduce some certain transformation constraint between individual and global actions. Both the theory and the experiments verify its effectiveness.
>
> Our method can benefit more for those scenarios that individual agent has partial observations, such as StarCraft Multi-Agent Challenge (SMAC). Due to the insufficient observation of global information, the optimal actions of individual agent are impossible to be equal to global optimal actions. Moreover, if taking action transformation, as plotted in Fig.1(c) in the original manuscript, the Q-value inconsistency between global and individual agents could be reduced. Furthermore, the experiments on SMAC demonstrate that our method can increase the average success rates/episode rewards.
>
> **Q2 (w.r.t Weakness 2): The experiments of this method are not convincing. More IGM methods such as WQMIX [NeurIPS 20], QPLEX [ICLR 21], etc. could be added to prove the effect of this work.**
>
> According to your suggestion, we conduct addition experiments to evaluate our framework (IGNT-MAC) compared with two baseline: Weighted-QMIX (**Rashid et al. (2020)**)  and QPLEX (**Wang et al.(2021)**) , on three scenarios of SMAC benchmark: 1c3s5z, 3s\_vs\_5z and MMM2. In particular, Weighted-QMIX proposed two scalable versions of algorithm, CW-QMIX and OW-QMIX.  We evaluate each method four times with different seeds and report the mean and standard deviation. We show the results in the following table. The experimental results shows that these methods integrated with our IGNT rule achieve better performance compared to the original baselines especially in Hard (3s\_vs\_5z) and Super-Hard (MMM2) scenarios.
>
> Table 1. The average success rates \% for the baselines and their variants under our framework.
> |      | OW-QMIX | OW-QMIX+IGNT(ours)   | QPLEX | QPLEX+IGNT(ours)     |
> | :---        |    :----:   |      :---: |       :----:   |      :---: |
> | 1c3s5z      | **88.36±4.12**      | 87.86±6.27   |  91.62±5.08      | **92.03±3.89**   |
> | 3s\_vs\_5z   |  16.62±3.04        | **33.29±2.38**      |   68.47±12.76      | **80.28±3.82**   |
> | MMM2   |  34.58±4.70       | **53.22±3.94**      |  53.41±20.46     | **71.83±4.91**   |
>
> **Q3 (w.r.t Weakness 3): This paper provides a proof of improvement for the policy-based method, however, the proof of the value decomposition method is not yet clear.**
>
> For the question you asked is related to **Lemma 2** (in this manuscript), which does not make the factorizable assumption on Q-value and thus is applicable to general settings. In other word, the derivation on the case of Q-value decomposition is a special case of Lemma 2. Here we provide a derivation example for monotonic linear value decomposition (MLVD): $Q_{\text{jt}}(\tau, u) = \sum_{i=1}^{N}{w_i(\tau_i) Q_i(\tau_i, u_i)}$. Other value decomposition ways could have the similar derivation. For more clear exhibition, we summarize a conclusion and the corresponding proof in **Appendix C.4**(Supplementary material).

---

> ### Author Response · Authors · 2023-11-22
> **Initial Response to Reviewer s6zd (Part 2/2)**
>
> **Q4 (w.r.t Question 1): Could you describe a scenario where the IGM or IGO conditions could lead to poor performance?**
>
> As shown in Figure 2 of the manuscript, we can observe that, in the super-hard scenario MMM2, the IGM constraint and IGO constraint (the last column) cannot achieve good performance. The scenario MMM2 is an asymmetric task that the quantity of enemy units is larger than the number of allied units, which requires MARL method to learn to combine these units to form powerful tactics and strategies, and then achieve victory in the battle.
>
> **Q5 (w.r.t Question 2): A closely related work is missing. ResQ [1] is a decomposition-based MARL method which learn a value decomposition by using a nonlinear function.**
>
> Thanks for your recommendation. This work ResQ is based on the IGM condition in essence, which is different from ours. We have cited the closely related work ResQ in the updated version.
>
> Reference
> [1] Siqi Shen, Mengwei Qiu, Jun Liu, Weiquan Liu, Yongquan Fu, Xinwang Liu, and Cheng Wang. Resq: A residual q function-based approach for multi-agent reinforcement learning value factorization. Advances in Neural Information Processing Systems, 35:5471–5483, 2022.
>
>
>
> **Q6 (w.r.t Question 3): In value-based decomposition MARL methods, do  Eq.(1)  and Eq.(2). satisfy IGM condition? Can the author provide the complete training procedure based on value decomposition such as QMIX?**
>
> Eq.(1): $Q_{jt}^{*} (\tilde{u} |\tau)$
>
> Eq.(2): $[Q_i^{*}( \tilde{u}_i|\tau)]$
>
> (i) The after-transfromed Eq.(1) and Eq.(2) usually do not satisfy the IGM condition. From the policy distribution perspective, the IGTO condition we propose does not constrain the individual-global action equivalency. As an extreme case that the normalized transformation is the Identity transformation,  Eq.(1) and Eq.(2) would satisfy the IGO condition, while IGM is a special case of IGO.
>
> (ii) According to your suggestion, here we give the description of the training procedure based on value decomposition such as QMIX.
>
> a) At the beginning of training, we initialize the environment and obtain the initial state and observations. b) Then, a action is selected from the individual policy according to the action-observation historical trajectory for each agent and the joint action, together formed by all agents, is transformed by the normalized transformation. c) Next, the transformed action is executed to obtain shared reward and the next observation. When the episode is terminated, the information of episode will be stored in replay buffer. d) After that, a batch of samples are sampled randomly from replay buffer and update each individual policy, the critic and the normalized transformation. e) After training every 200 steps, the individual target policy and critic will be updated.
>
> Note that these value-based composition methods need minor modifications to fit our IGNT-MAC framework. Concretely, for value-based composition method, we incorporate individual policy network and target network for each agent, and the original individual Q-value functions are retained to facilitate the learning of critic in the IGNT-MAC variant.
>
> **Q7 (w.r.t Question 4): For the discrete or continuous action space of an agent, how to ensure that all actions converted by the bijective function $f_i$ are valid, and they should include all actions in the raw action space.**
>
> Thanks for your comment. We make clarification that, in implementation, the transformation function is preformed in the continuous numerical space, i.e., its inputs and outputs are probability values. After transformation, the continuous probability values can be sampled to discrete and valid actions like plenty of single/multi-agent RL methods.
>
> **Please note that the updated manuscript for the revision has been uploaded.**

---

### Official Review · Reviewer_ujuj · 2023-11-05

**Soundness:** 2 fair
**Presentation:** 3 good
**Contribution:** 2 fair
**Rating:** 5
**Confidence:** 3

**Summary:**

The authors try to release the restriction that the optimal joint actions should
be consistent with the optimal individual behaviors. They propose the Individual-Global-Transform-Optimal condition, IGTO, which transforms the joint action via a bijection function. The authors claim that individual-global policies can converge to the optimum under this condition. The proposed method is easy to implement and can be integrated into many existing MARL methods seamlessly.

**Strengths:**

+ The paper is well-organized.
+ The experiments are extensive. IGNT achieves strong performance gains in many tasks compared with the backbone algorithm. The experiment settings and hyper-parameters are detailed. I think it is easy to reproduce the results.

**Weaknesses:**

First, is it really a restriction that the optimal joint actions should
be consistent with the optimal individual behaviors? I think it is a natural fact and can be satisfied in all environments. Can you propose a case (maybe matrix games) where the optimal joint action is inconsistent with the optimal individual actions, and perform experiments on it to show that your method can achieve the optimum as claimed?

The condition that is Jacobian determinant is 1 is derived by the change of variable formula. However, the formula requires that the Jacobian exists. In discrete action space, the function is not differentiable. Can you give us a numerical case with discrete action space to show the original actions, the transformed actions, the bijection function, and the Jacobian determinant? It would be helpful to understand your algorithm.

Moreover, without considering the Jacobian determinant, the function F is bijection. Does it really increase the representation abilities or release the restriction of individual actions? I cannot see the meaning of transforming the actions using a bijection function.

Considering the strong performance of IGNT, I would like to increase my score if the questions are addressed.

**Questions:**

See Weaknesses.

---

> ### Author Response · Authors · 2023-11-22
> **Initial Response to Reviewer ujuj (Part 1/2)**
>
> # Thank you for valuable comments. Below we respond to your concerns:
>
> **Question 1: is it really a restriction that the optimal joint actions should be consistent with the optimal individual behaviors? I think it is a natural fact and can be satisfied in all environments. Can you propose a case (maybe matrix games) where the optimal joint action is inconsistent with the optimal individual actions, and perform experiments on it to show that your method can achieve the optimum as claimed?**
>
> Thanks for your comment.
>
> (i) In fact, the restriction you said is unquestionable. In real world, the inconsistency between the optimal joint actions and the optimal individual behaviors often occurs if the agent cannot obtain global information (including other agents' states). This case might come from signal interruption, communication limitation or other factors such as in underwater, war battle environments. Hence, all CTDE methods (**Sunehag et al. (2018); Rashid et al. (2018); Son et al. (2019); Wang et al. (2020);
> Zhang et al. (2021); Wang et al. (2021)**) attempt to address this case, accordingly, we also do this.
>
> (ii) We conduct an additional experiment on the \textbf{Single-state Matrix Game}, a non-monotonic matrix game consists of two agents with three actions and a shared reward, to compare the variant under our framework (IGNT-MAC) and the original baseline. We take QMIX as the baseline. The results are shown in the following tables.
>
> As shown in Table 1, the joint optimal behavior is (A, A). For QMIX, as shown in Table 2, the individual optimal actions of agent 1 and 2 are B and C, respectively. In contrast, in Table 3, IGNT enables agents to reach the optimal joint behavior. It indicates that our method can reach the optimum. In addition, please also see a statistical comparison about the Q-value inconsistency between global and individual agents as plotted in Fig.1(c) in the original manuscript.
>
> Table 1. The payoff of matrix game.
> |   $u_1, u_2$   | A | B   | C    |
> | :---:       |    :----:   |      :---: |       :----:   |
> |A     | **8**      | -12   |  -12     |
> | B   |  -12        | 0    |   0     |
> | C   |  -12        | 0    |   0     |
>
> Table 2. QMIX: $Q_{jt}$.
> |   $u_1, u_2$   | A | B   | C    |
> | :---:       |    :----:   |      :---: |       :----:   |
> |A     | -8.1     | -8.1   |  -8.1     |
> | B   |  -8.1       | 0.0    |   **0.1**     |
> | C   |  -8.1        | 0.0    |   0.0     |
>
> Table 3. **QMIX+IGNT**: $Q_{jt}$.
> |   $u_1, u_2$   | A | B   | C    |
> | :---:       |    :----:   |      :---: |       :----:   |
> |A     | **8.1**     | -11.9  |  -11.8     |
> | B   |  -11.8       | 0.0    |   0.1     |
> | C   |  -11.8        | 0.1    |   0.0     |
>
> Reference
>
> [1] Peter Sunehag, Guy Lever, Audrunas Gruslys, Wojciech Marian Czarnecki, Vin ́ıcius Flores Zambaldi, Max Jaderberg, Marc Lanctot, Nicolas Sonnerat, Joel Z Leibo, Karl Tuyls, et al. Value-decomposition networks for cooperative multi-agent learning based on team reward. In AAMAS, 2018.
>
> [2] Tabish Rashid, Mikayel Samvelyan, Christian Schroeder, Gregory Farquhar, Jakob Foerster, and Shimon Whiteson. Qmix: Monotonic value function factorisation for deep multi-agent reinforcement learning. In International conference on machine learning, pp. 4295–4304. PMLR, 2018.
>
> [3] Kyunghwan Son, Daewoo Kim, Wan Ju Kang, David Earl Hostallero, and Yung Yi. Qtran: Learning to factorize with transformation for cooperative multi-agent reinforcement learning. In International conference on machine learning, pp. 5887–5896. PMLR, 2019.
>
> [4] Yihan Wang, Beining Han, Tonghan Wang, Heng Dong, and Chongjie Zhang. Off-policy multi-agent decomposed policy gradients. arXiv preprint arXiv:2007.12322, 2020.
>
> [5] Tianhao Zhang, Yueheng Li, Chen Wang, Guangming Xie, and Zongqing Lu. Fop: Factorizing optimal joint policy of maximum-entropy multi-agent reinforcement learning. In International Conference on Machine Learning, pp. 12491–12500. PMLR, 2021.
>
> [6] Jianhao Wang, Zhizhou Ren, Terry Liu, Yang Yu, and Chongjie Zhang. Qplex: Duplex dueling multi-agent q-learning. In ICLR, 2021.

---

> ### Author Response · Authors · 2023-11-22
> **Initial Response to Reviewer ujuj (Part 2/2)**
>
> **Question 2: The condition that is Jacobian determinant is 1 is derived by the change of variable formula. However, the formula requires that the Jacobian exists. In discrete action space, the function is not differentiable. Can you give us a numerical case with discrete action space to show the original actions, the transformed actions, the bijection function, and the Jacobian determinant? It would be helpful to understand your algorithm?**
>
> We first do the clarification that, in implementation, the transformation function is preformed in the continuous numerical space, i.e., its inputs and outputs are probability values.  That is, the transformation function is definitely differentiable, so the Jacobian matrix exists. Hence, the question, about the numerical case of the function in discrete space, makes no sense.
>
> The question you asked should be the case of taking discrete actions for environment interaction. Like plenty of single/multi-agent RL methods, the continuous probability values can be sampled to discrete actions. And, the differential computation of probability sampling has been implanted into many deep learning tools, such as pytorch we used, tensorflow, etc. We do not discuss the question here because it is beyond our scope.
>
> **Question 3: Moreover, without considering the Jacobian determinant, the function F is bijection. Does it really increase the representation abilities or release the restriction of individual actions?**
>
> The case you asked, not considering the Jacobian determinant equal to 1 but keeping the function bijection, would result into the  distribution inconsistency between joint optimal actions and individual optimal actions. Please refer to **the proof of Theorem 1**, given in **Part C of the supplementary material**.
>
> **Please note that the updated manuscript for the revision has been uploaded.**

---

> ### Comment · Reviewer_ujuj · 2023-11-22
>
> I see that in implementation the transformation function is preformed on the action probabilities. But in proof, u is a joint action, not action probability. So the proof seems to be wrong. Can you clarify the mismatch between proof and implementation?

---

> > ### Author Response · Authors · 2023-11-22
> >
> > Thanks for your timely feedback.
> >
> > We make sure there is no any problem. For example, we need select one action from three actions {A, B, C}. If A is definitely chosen, the action vector should be denoted as (1, 0, 0). But in practice, we need predict the action with the probability values like most deep policy-based reinforcement learning method. Suppose the prediction values are 0.85,0.05, 0.1 for A, B, C, the action vector will be (0,85, 0.05, 0,1). We use this type of probability vector as the input of the transformation. We hope that this could help you understand the question.
> >
> > Further, we strongly recommend you to read the code related to the topic of deep reinforcement learning.
> >
> > We sincerely appreciate your feedback again. If you have any other questions, please reply.

---

> ### Comment · Reviewer_ujuj · 2023-11-22
>
> The authors seem to think I am unfamiliar with deep RL.
>
> My concern is the theoretical analysis, which is independent from implementation. First, **Theorem 1 requires the Jacobian of discrete actions, so the Theorem is wrong.** Second, if the Theorem holds when replacing actions with action probabilities, you should re-write the proof because the proof is not trivial. Does Theorem 1 hold if u is action probability?

---

> > ### Author Response · Authors · 2023-11-22
> > **Response to Reviewer ujuj**
> >
> > Thanks for your timely reply.
> >
> > For clear explanation, we copy Theorem 1 to here:
> >
> >    **(Policy Preservation)** If we sequentially perform the transformation $f_i$: $[\widetilde{u}\_i; \widehat{u}\_{-i} ] =f\_i (u\_i, \widehat{u}\_{-i}),  \widehat{u}\_{-i} = [ \widetilde{u}\_1, \cdots, \widetilde{u}\_{i-1}; u\_{i+1}, \cdots, u\_{N} ]$, and the Jacobian determinant satisfies $|G\_i| =|\frac{\partial f\_i}{\partial[ u\_i; \widehat{u}\_{-i} ]} | = 1$, then individual global transform optimal in Definition 1  is provable.
> >
> > We first explain the symbols:
> >
> > i) $u_i, \widetilde{u}_i$ are the action vectors (please see the example just replied) of the $i$-th agent; $u_i$ is before transformation, and $\widetilde{u}_i$ is after transformation;
> >
> > ii) $\widehat{u}_{-i}$ is the concatenated vector from all agents except the $i$-th agent.
> >
> > ii) The symbol [~] denotes the concatenation of vectors;
> >
> > For convenient statement, we introduce the vector symbol $x,y$: $x=[u\_i, \widehat{u}\_{-i}]$ and $y=[\widehat{u}\_i, \widehat{u}\_{-i}]$. So the input and output of the transformation are $x$ and $y$ respectively.
> >
> > Suppose the action vector of each agent is with $d$ dimensions, then the input of the transforamtion $f_i$ is $x\in\mathbb{R}^{dN}$, where $N$ is the number of agents. And the output $y$ of the transformation is also a vector with $dN$ dimension. Hence, the Jacobian matrix is the gradient of $y$ w.r.t the input action vector $x$. It is the general Jacobian style. Hence, whether one-hot vector or probability vector for discrete actions, Theroem 1 holds. Accordingly, Theroem 1 and the proof do not have the problem you said.
> >
> > Please note that discrete actions are represented with vectors.
> >
> > We hope you can understand the answer. Thanks again.

---

> ### Comment · Reviewer_ujuj · 2023-11-22
>
> The gradient of $y$ w.r.t the input action vector $x$ is ill-defined if $x$ is discrete, whether it is represented with vectors or not.
>
> For example, $x \in [0,1,2,3], f = x^2$, $f'(2)$ does not exist. $x \in [(0,0,0,1),(0,0,1,0),(0,1,0,0),(1,0,0,0)], f = xx^T$, $f'((0,1,0,0))$ does not exist.
>
> Let us check where the Jacobian comes from. It is derived from the lines below Eq. 30, the change of variable formula. However, this technique can only be applied in continues variable. In discrete variable, the Jacobian cannot be defined. And we should use $\sum$ to deal with the change of discrete variable.

---

> > ### Author Response · Authors · 2023-11-22
> > **Response to Reviewer ujuj**
> >
> > Thanks for your timely reply.
> >
> > First, our implementation of the transformation is in the continuous value space, which is consistent with Theorem 1 in the continuous case.
> >
> > For the question whether Theorem 1 supports the discrete case, if existing Jacobian matrix, it should be provable.
> >
> > If Jacobian is not existed, you give an example. The example might not be proper here, because the values in Jacobian matrix don’t indicate the action and may surpass the scope. For the example you said, f’((0,1,0,0))=2*(0,1,0,0)=(0,2,0,0). f’ is the gradient not the action, and thus (0,2,0,0) is meaningful. In another view, if the function $f$ is differentiable, e.g., $x^2$, the gradient at one point can be given. Of course, the Jacobian matrix might be not existed in some cases. For more precision, we add the existing condition of Jacobian, which is update the manuscript.
> >
> > Thanks for you attention.

---

### Official Review · Reviewer_H7GA · 2023-11-06

**Soundness:** 3 good
**Presentation:** 3 good
**Contribution:** 3 good
**Rating:** 6
**Confidence:** 4

**Summary:**

This paper proposes an Individual-Global Normalized Transformation (IGNT) rule that maps a sample from a simple density i.e., Gaussian policy. In this way, the restriction of IGO is released and the global value function is able to represent complex situations. Experimental results show that the proposed method outperforms many state of the art baselines.

**Strengths:**

See questions

**Weaknesses:**

See questions

**Questions:**

This paper provides a simple but powerful method to release the restriction of IGO. By mapping the action to a distribution, the global value network can perform much better than multiple baselines. Theoretical analysis is also sound and provides the guarantee of convergence. The paper is interesting and easy to follow. However, I still have some concerns:

1. The proposed method is similar to the normalization of actions, which is widely used in many methods as a trick to improve performance. What is the difference between the proposed method and the widely-used normalization?

2. In the paper, the soft objective function is adopted. It seems that the proposed method can also apply to the normal objective function. Could authors explain why they must be combined? Since there is no ablation study to show the performance without the soft objective function, it is hard to say why the performance improved.


typos: a Individual-Global Normalized Transformation (IGNT) rule that map ..... -> an .... that maps .....;

---

> ### Author Response · Authors · 2023-11-22
> **Initial Response to Reviewer H7GA**
>
> # Thank you for recognizing the interesting work and proposing some valuable comments, below we make the responses to your concerns:
>
> **Question 1: The proposed method is similar to the normalization of actions, which is widely used in many methods as a trick to improve performance. What is the difference between the proposed method and the widely-used normalization?**
>
> Thanks for your comment. We clarify it from three points:
>
> (i) The purpose of this work is to relax the constraint of rigorous equivalency of individual-global actions in CTDE for boosting multi-agent learning. To this end, we propose a new condition, called individual-global action-transformed (IGTO), which requires that the Jacobian determinant of transformation is equal to one. We theoretically prove that the convergency and optimality of multi-agent learning can be well guaranteed, when this condition is hold.
>
> (ii) As an alternative of solving the requirement, i.e., satisfying this new condition, we propose an individual-global normalized transformation (IGNT) way by constraining the continuous transformation from global actions to local actions in a normalized mode as called in the paper. In fact, other transformation modes may be employed only if satisfying this new condition. In other word, *the normalized transformation (even literally like the normalization you said) is an only alternative way to implement our method.*
>
> (iii) The normalization you said usually constrains Jacobian matrix as some special matrices (e.g., upper/lower triangular matrix) in order for high-coefficient computation or other requirements. In contrast, the normalization used in the work is to *impose the scale constraint on Jacobian matrix* w.r.t the value of Jacobian determinant. Of course, the widely-used normalization you said may be used in our work as a way to accelerate and improve our algorithm.
>
> **Question 2: In this paper, the soft objective function is adopted. It seems that the proposed method can also apply to the normal objective function. Could you explain why they must be combined?**
>
> Thanks for this comment. The soft objective function, you said, is widely-used as an entropy regularization term for single/multi-agent reinforcement learning (RL). The advantages of combining the soft objective term are two folds: a) increase action exploration space  (**Grau-Moya et al. (2016)**); b) reducing value overestimation  (**Fox et al. (2015)**). In practice, we observe that this term indeed improve the robustness of algorithm as declared in many previous methods (**Fox et al. (2015), Grau-Moya et al. (2016), Haarnoja et al. (2018)**).
> Hence, we also introduce it as a regularization in our method as well as all comparisons in the manuscript. Please note that, in our experiment, the soft objective term are incorporated into all compared methods for fair comparisons, where the original FOP itself has adopted the term.
>
> Reference
>
> [1] Roy Fox, Ari Pakman, and Naftali Tishby. Taming the noise in reinforcement learning via soft updates. arXiv preprint arXiv:1512.08562, 2015.
>
> [2] Jordi Grau-Moya, Felix Leibfried, Tim Genewein, and Daniel A Braun. Planning with information-processing constraints and model uncertainty in markov decision processes. In Machine Learning and Knowledge Discovery in Databases: European Conference, ECML PKDD 2016, Riva del Garda, Italy, September 19-23, 2016, Proceedings, Part II 16, pp. 475–491. Springer, 2016
>
> [3] Tuomas Haarnoja, Aurick Zhou, Pieter Abbeel, and Sergey Levine. Soft actor-critic: Off-policy maximum entropy deep reinforcement learning with a stochastic actor. In International conference on machine learning, pp. 1861–1870. PMLR, 2018
>
> **Question 3: There is a typos in this paper: a Individual-Global Normalized Transformation (IGNT) rule that map.**
>
> Thanks for your careful review. We have polished this manuscript by correcting typos (including that you said) and grammar errors in the updated version.
>
> **Please note that the updated manuscript for the revision has been uploaded.**

---

### Meta-Review · Area_Chair_FRAh · 2023-12-06

**Metareview:**

The paper proposes Individual-Global Transform-Optimal (IGTO) that allows inconsistent individual-global actions while ensuring the equivalency of their policy distributions for Centralized Training with Decentralized Execution (CTDE) in Multi-Agent Reinforcement Learning (MARL). Based on IGTO, Individual-Global Normalized Transformation (IGNT) is further proposed, which can be integrated with many existing CTDE methods. The experiments show improvement over base methods.

After reading the paper, the reviews, and the authors' responses, I think there are a few concerns raised by the reviewers, which are not addressed during the rebuttal.

- The main concern is the theory. First, adding the condition that "if the Jacobian matrix of the transformation exists" does make Theorem 1 hold for discrete actions. In theory, the Jacobian matrix does not exist for discrete actions. Thus, the paper should make it clear that Theorem 1 does not hold for discrete action. Second, during the rebuttal, the authors gave an example of discrete actions (0.85, 0.05, 0.1), which, however, is the categorical distribution of the discrete actions. It seems the authors mixed up the concept of discrete actions in theory and the categorical distribution of the discrete actions in implementation. If Theorem 1 is provable for the categorical distribution of the discrete actions, this should be included in the paper. As the main experiments are on SMAC that has discrete action space, the proofs for discrete actions/the categorical distribution of the discrete actions are necessary.

- The motivation is not well supported. During the rebuttal, the authors give an example. However, such a matrix game can be well solved by existing methods, like QPLEX (IGM) and FOP (IGO).

- It is unclear how IGNT is adapted to value-based decomposition CTDE MARL methods. The authors do not provide any useful information during the rebuttal. Since IGNT is combined with VDN, QMIX, and QTRAN in the experiments, this should be made clear and included in the paper.

- As IGNT is an actor-critic framework, it should be well evaluated based on actor-critic methods. However, currently, the majority of the evaluation is compared to value-based methods, like VDN, QMIX, and QTRAN. The authors mentioned that the soft objective term is incorporated into all compared methods for fair comparisons. However, the induced QMIX seems to perform worse than the original QMIX, like on the map MMM2. Indeed, more actor-critic baselines should be included.

**Justification For Why Not Higher Score:**

The concerns mentioned above are not addressed during the rebuttal, which makes the proposed IGTO and IGNT inconclusive. Thus, this paper is not ready for publication.

**Justification For Why Not Lower Score:**

N/A

---

### Decision · Program_Chairs · 2024-01-16

Reject